# VenusBench-Mobile: A Challenging and User-Centric Benchmark for Mobile GUI Agents with Capability Diagnostics

Yichen Gong [* 1]  Zhuohan Cai [* 2 1]  Sunhao Dai [1]  Yuqi Zhou [1]  Zhangxuan Gu [1]  Changhua Meng [1]
Shuheng Shen [1]

## Abstract

Existing online benchmarks for mobile GUI agents remain largely app-centric and task-homogeneous, failing to reflect the diversity and instability of real-world mobile usage. To this end, we introduce VenusBench-Mobile, a challenging online benchmark for evaluating general-purpose mobile GUI agents under realistic, user-centric conditions. VenusBench-Mobile builds two core evaluation pillars: defining what to evaluate via user-intent-driven task design that reflects real mobile usage, and how to evaluate through a capability-oriented annotation scheme for fine-grained agent behavior analysis. Extensive evaluation of state-of-the-art mobile GUI agents reveals large performance gaps relative to prior benchmarks, indicating that VenusBench-Mobile poses substantially more challenging and realistic tasks and that current agents remain far from reliable real-world deployment. Diagnostic analysis further shows that failures are dominated by deficiencies in perception and memory, which are largely obscured by coarse-grained evaluations. Moreover, even the strongest agents exhibit near-zero success under environment variations, highlighting their brittleness in realistic settings. Based on these insights, we believe VenusBench-Mobile provides an important stepping stone toward robust real-world deployment of mobile GUI agents. Code and data are available at https://github.com/inclusionAI/UI-Venus/tree/VenusBench-Mobile.

*Equal contribution [1]Venus Team, Ant Group, Hangzhou, Zhejiang, China [2]Department of Computer Science and Technology, Tsinghua University, Beijing, China. Correspondence to: Yichen Gong <gongyichen.gyc@antgroup.com>, Shuheng Shen <shuheng.ssh@antgroup.com>.

*Proceedings of the 43rd International Conference on Machine Learning*, Seoul, South Korea. PMLR 306, 2026. Copyright 2026 by the author(s).

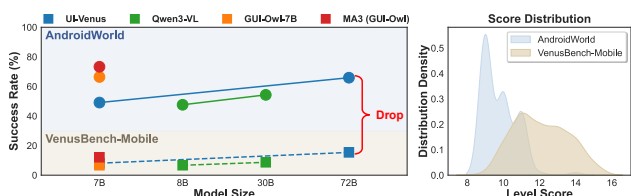

*Figure 1.* Overall comparison between AndroidWorld and VenusBench-Mobile. **Left:** Agent performance on VenusBench-Mobile and AndroidWorld. **Right:** Distribution of task difficulty measured by the capability level score. A higher score indicates higher requirements for the agent.

## 1. Introduction

The rapid progress of visual language models (VLMs) (Hurst et al., 2024; Bai et al., 2025; Hong et al., 2025) is enabling agents to interact with mobile graphical user interfaces (GUIs) through visual perception and natural-language instructions. Typically, mobile GUI agents can interpret on-screen content, plan multi-step workflows, and execute actions over multiple turns (Gu et al., 2025; Qin et al., 2025; Xu et al., 2025; Ye et al., 2025; Yan et al., 2025), making them a compelling direction for improving everyday mobile experiences. Recent commercial systems such as Doubao AI Smartphone (ByteDance, 2025) and AutoGLM (Liu et al., 2024) further suggest practical demand, demonstrating end-to-end task automation in scenarios such as cross-app price comparison and multi-platform travel booking. These developments elevate mobile GUI agents from single-app automation tools to general-purpose mobile assistance, which in turn makes rigorous evaluation increasingly critical.

To evaluate such agents, dynamic simulated-environment benchmarks (Rawles et al., 2024; Guo et al., 2025), often referred to as **online benchmarks**, have become the dominant methodology. Unlike static datasets, representative online benchmarks such as AndroidWorld (Rawles et al., 2024) provide controllable and reproducible environments in which agents can execute sequential actions, observe state transitions, and receive real-time feedback, offering the closest approximation to real-world deployment conditions. Despite these advantages, existing online mobile GUI

benchmarks still exhibit two critical limitations that hinder progress toward robust general-purpose mobile assistants.

**First, current evaluations do not align well with real-world user requirements.** Most benchmarks follow a **bottom-up**, *app-centric* construction process: they curate a set of apps and then create tasks that target app-specific functionalities. This design implicitly treats apps as evaluation targets, whereas a general-purpose assistant should treat apps as tools to satisfy diverse user intents in the broader phone environment. As a result, the induced task distribution is often narrow and homogeneous, underrepresenting compositional, cross-app, and intent-driven behaviors that matter in practice. This mismatch raises a central concern: strong performance on existing benchmarks may not translate to reliable real-world usefulness.

**Second, existing benchmarks provide limited diagnostic power for understanding and improving agent capabilities.** They typically report coarse success rates with little systematic attribution of failures to underlying causes. When an agent fails, it is often unclear whether the bottleneck lies in visual perception, state tracking and memory, decision-making, or another capability. Moreover, as leading models approach near-ceiling scores on some benchmarks, aggregate success rates become less discriminative, making it difficult to distinguish strong agents from marginally better ones or to identify actionable directions for improvement.

To address these limitations, we introduce VenusBench-Mobile, a challenging online benchmark for evaluating general-purpose mobile GUI agents under realistic, *user-centric* conditions. VenusBench-Mobile is designed to reflect how mobile assistants are used in practice from a **top-down** view, rather than how individual app functionalities are exercised in isolation. Specifically, VenusBench-Mobile adopts a user-intent–driven task design that captures a broad spectrum of real-world mobile usage beyond single-app workflows, spanning ten major categories of user intents. To better reflect deployment-time complexity, we further introduce systematic environment variations, such as changes in language, layout, and interface conditions, to assess agent robustness and behavioral stability under realistic distribution shifts. Moreover, to enable fine-grained diagnosis of agent failures, each task is annotated with a capability-oriented diagnostic scheme that characterizes the underlying abilities required for successful completion, allowing failures to be attributed to specific capability deficiencies rather than reported as aggregate success rates.

Based on VenusBench-Mobile, we conduct extensive evaluations of state-of-the-art (SOTA) mobile GUI agents and obtain several key insights. Compared to prior benchmarks, SOTA agents exhibit large performance drops, with average success rates decreasing by around 50 points, indicating that VenusBench-Mobile poses substantially more challenging

and realistic tasks and that current agents remain far from reliable real-world deployment. Diagnostic analysis further reveals that failures are dominated by deficiencies in perception and memory, limitations that are largely obscured by coarse-grained evaluations. Moreover, even the strongest agents achieve near-zero success under environment variations, exposing their brittleness to realistic distribution shifts. Taken together, these results suggest that robust real-world mobile assistance remains an open challenge. Based on these insights, we believe VenusBench-Mobile serves as a foundational evaluation benchmark, providing essential guidance for advancing mobile GUI agents toward more general, robust, and deployable assistants.

In summary, our main contributions are as follows:

• We introduce VenusBench-Mobile, a challenging online benchmark with a user-intent–driven task design that better reflects realistic mobile usage beyond app-centric and homogeneous task settings.

• We propose a capability-oriented diagnostic annotation scheme that enables fine-grained analysis of agent failures and distinguishes underlying capability deficiencies that are obscured by coarse-grained success metrics.

• Extensive evaluation provides insights into the limitations of current mobile GUI agents, showing that SOTA agents remain far from reliable real-world deployment due to bottlenecks in perception, memory, and robustness under realistic environment variations.

**Conflict of Interest Disclosure.** Several authors are affiliated with Ant Group, and this work evaluates UI-Venus models developed by Ant Group/Venus Team. The authors disclose this relationship as a potential financial conflict of interest.

## 2. Related Work

**Mobile GUI Agents.** Mobile GUI agents represent an emerging paradigm in human-computer interaction, evolving from early rule-based scripts to autonomous systems powered by foundation models. These agents can be categorized by their input modalities, ranging from text-only instructions to multimodal inputs combining text and screenshots, or even pure-visual perception for cross-app automation (Gu et al., 2025; Qin et al., 2025; Ye et al., 2025; Zhou et al., 2025b). In terms of architecture, current research differentiates between modular agent frameworks built upon Large Foundation Models and specialized end-to-end models optimized for GUI grounding(Yan et al., 2025; Zhou et al., 2025a). Despite these advancements, the inherent complexity of mobile environments poses significant challenges to agent reliability and execution consistency. Consequently, establishing robust evaluation frameworks

*Table 1.* A comparative overview of online mobile GUI benchmarks, highlighting VenusBench-Mobile's unique focus on user-centric requirements. "Verif." indicates whether the benchmark supports automated verification of task completion, enabling reproducible evaluation. "Cost" denotes whether the benchmark reports inference costs during task execution, such as time or token consumption.

| Benchmark | # Apps | # Tasks | Verif. | Cost | Task Categories | | | | | | | | | |
|---|---|---|---|---|---|---|---|---|---|---|---|---|---|---|
| | | | | | FA | CF | VA | MR | GSA | GUIM | HGB | NR | BC | ST |
| LearnGUI (Liu et al., 2025) | 20 | 101 | ✓ | | | | | | | | ✓ | | | |
| MMBench-GUI (Wang et al., 2025) | - | 146 | ✓ | | | | | | | | | | | |
| UI-NEXUS (Guo et al., 2025) | 20 | 100 | ✓ | ✓ | ✓ | | | | | | | | | |
| MVISU (Huang et al., 2025) | 137 | 404 | | ✓ | | ✓ | ✓ | | | | ✓ | | | |
| AndroidWorld (Rawles et al., 2024) | 20 | 116 | ✓ | ✓ | | | | | | | ✓ | | | ✓ |
| AndroidLab (Xu et al., 2024) | 9 | 138 | ✓ | ✓ | | | | | | | | | | |
| MobileAgentBench (Wang et al., 2024) | 10 | 100 | ✓ | ✓ | | | | | | | | | | |
| AndroidDaily (Team, 2025) | 48 | 235 | | | | | | | | | ✓ | | | |
| SPABench (Chen et al., 2025) | 66 | 340 | | ✓ | | | | | | | | | | |
| MobileWorld (GUI) (Kong et al., 2025) | 20 | 161 | ✓ | | | ✓ | ✓ | ✓ | | | | | | |
| VenusBench-Mobile (**Ours**) | 27 | 149(+80) | ✓ | ✓ | ✓ | ✓ | ✓ | ✓ | ✓ | ✓ | ✓ | ✓ | ✓ | ✓ |

is essential to benchmark current progress and guide the development of more capable general-purpose assistants.

**Benchmarks for Mobile GUI Agents.** The evaluation of GUI agents has progressed across offline and online paradigms. *Offline benchmarks*, such as AndroidControl (Li et al., 2024) and GUIOdyssey (Lu et al., 2025), use static datasets to evaluate action prediction but lack the closed-loop feedback required for live system assessment. In contrast, *online benchmarks* like AndroidWorld (Rawles et al., 2024) provide interactive environments for verifiable task execution. Subsequent works, including SPA-Bench (Chen et al., 2025) and MobileWorld (Kong et al., 2025), have scaled these environments or introduced hybrid paradigms like tool invocation via the Model Context Protocol (MCP). Nevertheless, existing benchmarks remain predominantly app-centric, focusing on functional completion within static contexts. They often overlook the volatile nature of user intents and lack the cognitive depth required for sophisticated GUI reasoning. To address these limitations, we propose VenusBench-Mobile, which shifts the evaluation paradigm from basic functional completion to a user-centric perspective.

## 3. VenusBench-Mobile Benchmark

This section introduces the design and construction of VenusBench-Mobile. Our benchmark is built around two core questions: **what** mobile agent tasks should be evaluated, and **how** agent performance should be assessed.

Unlike existing benchmarks that define tasks primarily based on app functionalities, VenusBench-Mobile adopts a **top-down, user-intent–driven design**. We view mobile GUI agents as general-purpose assistants deployed in real-world scenarios, where users express diverse and often imperfect intents. Accordingly, our tasks are designed from a **user-centric perspective**, covering a wide range of interaction patterns, including functional assistance, conflict resolution, vague or underspecified instructions, and multi-round interactions (Sec. 3.1).

Beyond task success, we further evaluate agents through a structured **capability evaluation framework**. Each task is annotated with multiple capability dimensions, enabling fine-grained analysis of agent behavior and failure modes. This design allows us not only to measure overall performance, but also to diagnose which specific abilities limit task success under different conditions (Sec. 3.2).

### 3.1. User-Intent-Driven Task Categories

Before curating specific tasks, we identify fundamental user needs in mobile environments, resulting in 10 task categories. Figure 2 illustrates these categories with representative task examples and expected agent behaviors.

**Function Assistance (FA).** Existing benchmarks assume users already possess complete knowledge about the functionalities each app provides and the procedures to access them. In reality, users often have incomplete or uncertain understanding of app capabilities. This category encompasses three types of tasks: Operation Explanation, where agents explore apps and report procedural knowledge for accomplishing specific tasks (e.g., *"Please tell me how to use ZipXtract to extract files"*); App Capability Discovery, where agents investigate app features and determine whether specific functionality exists (e.g., *"I want to create a new LaTeX file in Markor. Can I create a LaTeX file and compile it to a PDF?"*); and Interface Navigation, where agents help locate relevant interfaces that align with user goals (e.g., *"Explore Fitbook and show me the interface where I can add a new weight record"*).

**Conflict (CF).** Agents must identify and resolve conflicts or ambiguities between user instructions and the actual GUI

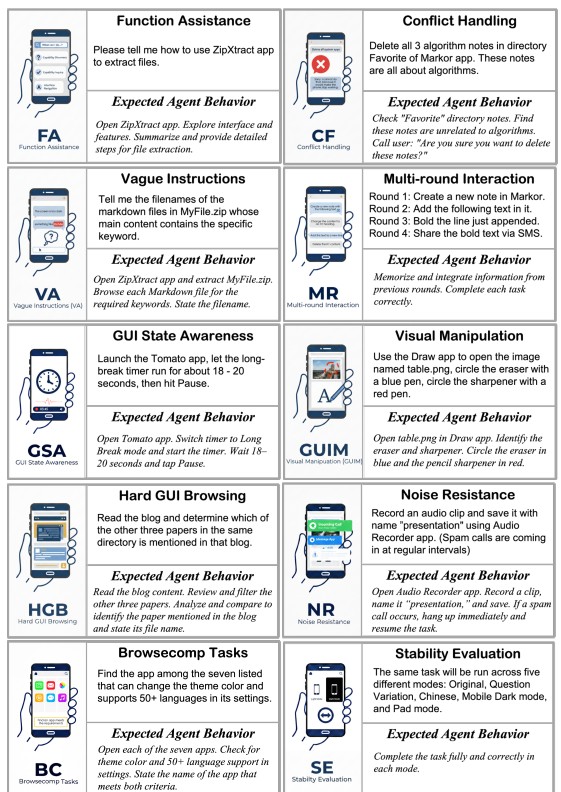

Figure 2. Taxonomy of task categories and illustrative examples in VenusBench-Mobile.

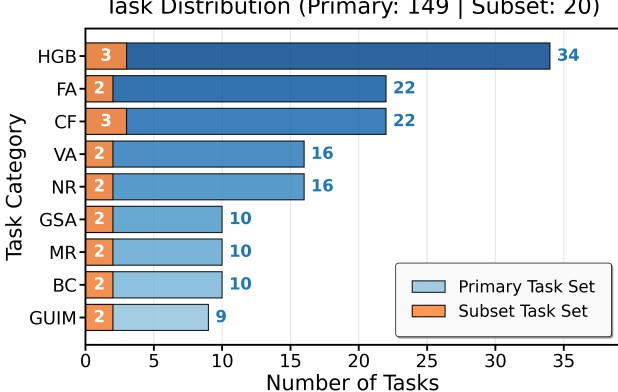

Figure 3. Distribution of task categories in VenusBench-Mobile, comprising a primary set of 149 tasks and a stability evaluation subset of 20 tasks.

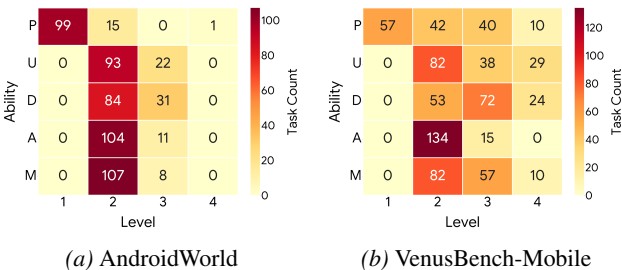

*(a) AndroidWorld*    *(b) VenusBench-Mobile*

Figure 4. Heatmap of task distribution across the five PUDAM dimensions and four difficulty levels. Each cell represents the number of tasks requiring a specific level for each dimension.

environment. When conflicts arise, agents should prompt users to respecify their requirements. For example, given the instruction *"Delete all 3 notes in Algo directory of Markor"* but this directory contains 4 notes, an agent should detect such conflict and seek user clarification on how to proceed.

**Vague Instructions (VA).** Real users often provide underspecified instructions that omit key contextual information. In these tasks, users have clear intents but express them without explicit details, such as omitting app names (e.g., wanting to extract a zip file without mentioning ZipXtract).

**Multi-round Interaction (MR).** GUI agents should support multi-round dialogues where users iteratively refine or redirect tasks. This includes: **(1) Sequential refinement**—users modify completed results with new requests (e.g., "change the title" after creating a note), and **(2) Mid-execution interruption**—users interrupt ongoing tasks with immediate new requests (e.g., "delete this note" while it's being created).

**GUI State Awareness (GSA).** Unlike existing benchmarks where information is pre-initialized, real-world mobile GUIs are dynamic. This category requires agents to continuously monitor evolving states and make decisions based on newly appearing information, such as processing incoming messages by specified rules or recording audio for durations shown in real-time display.

**Visual GUI Manipulation (GUIM).** This category represents user needs for GUI agents to perform operations on visual elements within the mobile GUI interface. Tasks involve precise visual creation and modification, requiring fine-grained coordinate-level control and strong visual element localization capabilities. Examples include drawing specific shapes or characters (e.g., "draw the letter A") and marking requested content in images (e.g., "circle the banana with a red pen").

**Hard GUI Browsing (HGB).** Users frequently need to gather and compare information across multiple sources on mobile devices. Unlike simple search-based retrieval, these tasks require extensive page browsing, information gathering, and processing across mobile interfaces. Compared to AndroidWorld's Information Retrieval tasks, our HGB tasks are significantly more demanding in two key aspects: **(1) Information volume**, requiring agents to process substantially more content across multiple screens and apps, and **(2) Reasoning complexity**, demanding synthesis and comparative analysis rather than simple information extraction. Browsing objects span diverse modalities including videos, PDFs, images, non-searchable app GUIs, and HTML pages.

*Table 2.* Task variants in the Stability Evaluation subset. Each original task is augmented with four systematic variants, yielding 80 additional instances.

| Variant | Description | Instances |
|---|---|---|
| Original | Original task | 20 |
| Chinese | Chinese translation of instructions | 20 |
| Variation | Semantically equivalent English instructions | 20 |
| Dark | GUI rendered in dark theme | 20 |
| Pad | Tablet (2560×1600) in landscape mode | 20 |
| **Total** | - | 100 |

**Noise Resistance (NR).** In real-world mobile usage scenarios, users inevitably encounter environmental disruptions during task execution, such as incoming calls, app crashes, and unrelated notifications. This category evaluates agents' in-task robustness—the ability to handle these interruptions and successfully recover to complete the original task. We simulate four types of environmental noise: incoming calls, app crashes, operation failures, and unrelated app popups. The underlying tasks are adapted from AndroidWorld (Rawles et al., 2024), with noise injected.

**Browsecomp-like (BC).** While the above categories cover common user scenarios, real-world usage inevitably includes corner cases—highly challenging tasks requiring comprehensive capabilities of GUI agents. Inspired by BrowseComp (Wei et al., 2025), a benchmark designed to evaluate deep-research agents on complex web browsing tasks with multiple constraint conditions, we adapt this paradigm to mobile GUI environments. Our BC tasks similarly incorporate multi constraints within mobile contexts. For example, *"find an app that has a blue icon and whose main interface does not display a search bar."*

**Stability Evaluation (SE).** Real-world users require agents to reliably handle similar tasks across varying conditions, as a single failure among repeated executions degrades user experience. To capture this aspect, we introduce Stability Evaluation (SE), which measures **consistency**—whether agents can solve the *same* task under diverse conditions such as paraphrased instructions or different GUI settings. For this purpose, we construct a dedicated SE subset: for each selected representative task, we generate multiple systematic variants to test agents' robustness to linguistic and visual perturbations. This design enables comprehensive assessment of agent stability across different instruction formats and interface presentations.

### 3.2. Capability Taxonomy for Mobile GUI Agents

To enable systematic evaluation of GUI agent competencies, we propose a comprehensive capability framework that decomposes agent abilities into five fundamental dimensions.

**Perception (P)** measures the agent's ability to understand and extract information from the GUI environment. This encompasses recognizing UI elements, comprehending layout structures, achieving high-precision localization, and perceiving dynamic changes.

**Understanding (U)** evaluates the agent's ability to comprehend user instructions. This ranges from parsing explicit commands to handling complex conditional constraints and resolving ambiguous or implicit intents.

**Decision (D)** assesses the agent's strategic reasoning during task execution, progressing from deterministic path following to dynamic strategy adaptation under environmental uncertainty, and ultimately to reflective planning with error correction and risk assessment.

**Action (A)** evaluates the agent's operational capabilities in executing GUI interactions. This encompasses basic touch operations, complex trajectory control, fine-grained semantic operations requiring spatial accuracy, and real-time closed-loop manipulation.

**Memory (M)** measures the agent's ability to retain and utilize task-relevant information throughout execution. This includes maintaining context across screen transitions, accumulating information for task completion (e.g., browsing multiple pages to gather data), and tracking multi-round dialogue history.

Each dimension comprises four progressive proficiency levels, detailed in Appendix A.

### 3.3. Evaluation Infrastructure

**Platform Selection.** To instantiate our task categories into an executable benchmark with reproducible results, we build upon AndroidWorld (Rawles et al., 2024), currently the most widely-used dynamic mobile GUI evaluation platform. This framework utilizes Android emulators as interactive GUI environments and leverages OS state inspection for task success verification. Building on this established infrastructure reduces the development cost for evaluating new agents on our benchmark.

**Application Selection.** Our benchmark incorporates 27 open-source Android apps: the 20 apps from AndroidWorld's original suite, plus 7 additional apps we introduce to enhance GUI diversity and coverage of usage scenarios. Details of the 7 newly added apps are provided in Table 6.

**Task Curation.** Based on our task taxonomy and selected apps, we manually construct tasks spanning nine categories (excluding SE), yielding a primary pool of 149 tasks. Figure 3 shows the distribution across nine categories. We also curate task variations for SE, with details and instance counts summarized in Table 2. More details about task curation and setting are provided in Appendix B.1.

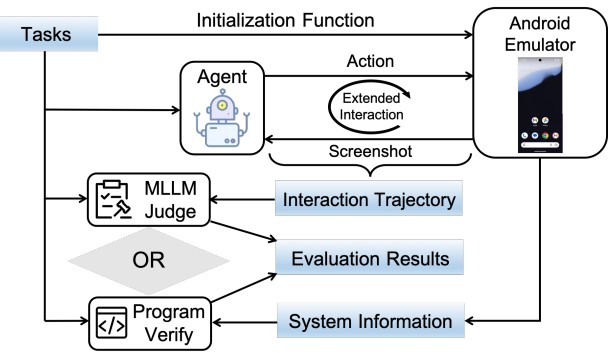

*Figure 5.* Overall execution flow and evaluation infrastructure of VenusBench-Mobile, illustrating the extended interactions (multi-round execution and noise-resistance tasks) between the mobile agent and the Android emulator, as well as a hybrid verification pipeline that applies MLLM-as-a-Judge for semantic tasks and programmatic verification for state-based tasks.

**Framework Development.** As shown in Figure 5, we adopt AndroidWorld's core architecture and introduce several extensions to support our benchmark requirements. **(1) Reproducible Evaluation Infrastructure.** We implement task-specific **initialization and verification functions**. Initialization functions prepare consistent starting states (e.g., importing files, configuring app settings). For verification, tasks are evaluated through either programmatic OS state inspection or MLLM-based judgment, depending on task characteristics. For tasks requiring visual recognition or semantic understanding, we employ MLLM-based judgment (details in Appendix B.4). **(2) Advanced Task Scenario Support.** We extend the framework with: (a) dynamic noise injection mechanisms that trigger perturbations during execution, supporting the four noise types in the NR category; and (b) multi-round dialogue capabilities that enable agents to handle follow-up instructions and user clarifications, as required by the MR category.

## 4. Evaluation Results and Analysis

In this section, we present the experimental results of mainstream mobile GUI agents on VenusBench-Mobile. We analyze the performance from multiple dimensions: overall success rates, capability proficiency levels, stability under perturbations, and inference efficiency. The evaluation is conducted in a purely vision-based manner, where the GUI environment provides only screenshots to the agents as their sole perception input. For closed-source models such as Gemini-3-Pro and GPT-5.1, we utilize UI-Venus-72B as the *grounding executor*, while the closed-source models themselves function as the *planner*. Additional details are provided in Appendix D.

### 4.1. Metric

We comprehensively evaluate mobile GUI agents using two task pools with corresponding metrics:

**Primary Task Success Rate (SR).** For the primary task pool consisting of 149 tasks, we measure the standard Success Rate (**SR**):

$$\text{SR} = \frac{1}{N} \sum_{i=1}^{N} y_i, \tag{1}$$

where $N$ is the total number of primary tasks, and $y_i \in \{0, 1\}$ indicates whether task $i$ was successfully completed.

**Dimension-wise Accuracy ( $Acc_{dim}$).** To evaluate specific competencies, we calculate the accuracy for each dimension $d \in \{P, U, D, A, M\}$ at different difficulty tiers:

$$Acc_{dim}(d, Tier) = \frac{Y_{d,Tier}}{X_{d,Tier}}, \tag{2}$$

where $X_{d,Tier}$ represents the total number of tasks requiring capability $d$ at the designated levels within a tier (e.g., L1&2 or L3&4), and $Y_{d,Tier}$ is the number of those tasks successfully completed by the model.

**Stability Evaluation.** To evaluate agent consistency under varying conditions, we measure performance on the task variants summarized in Table 2 using a *Stability Pass Rate* (**SPR**). SPR reflects the fraction of base tasks for which an agent successfully completes *all* corresponding variants:

$$\text{SPR} = \frac{1}{M} \sum_{j=1}^{M} \mathbb{1} \left( \sum_{k=1}^{K_j} y_{jk} = K_j \right), \tag{3}$$

where $M$ is the number of base tasks, $K_j$ is the number of variants for base task $j$, $y_{jk} \in 0, 1$ indicates whether the $k$-th variant was successfully completed, and $\mathbb{1}(\cdot)$ outputs 1 only if all $K_j$ variants succeed.

**Inference Cost.** To quantify the computational and economic overhead of mobile GUI agents, we examine the output token consumption. We prioritize token counts over execution time as they provide a hardware-independent measure of an agent's generative reasoning density and directly reflect API billing costs. We define two metrics:

**Total Tokens (TT)** represents the cumulative number of output tokens generated across all tasks:

$$\text{TT} = \sum_{i=1}^{N} \sum_{t=1}^{T_i} tokens_{i,t}^{out}, \tag{4}$$

where $N$ is the total number of tasks, $T_i$ is the number of steps taken for task $i$, and $tokens_{i,t}^{out}$ denotes the number of output tokens generated at step $t$.

*Table 3.* Success Rate (%) of different mobile GUI agents on VenusBench-Mobile.

| Model | FA | CF | VA | MR | GSA | GUIM | HGB | NR | BC | Total |
|---|---|---|---|---|---|---|---|---|---|---|
| *Open-source* | | | | | | | | | | |
| UI-Venus-72B | 22.7 | 4.6 | 12.5 | 0.0 | 10.0 | 0.0 | 17.7 | 50.0 | 0.0 | 15.4 |
| UI-Venus-7B | 13.6 | 4.6 | 25.0 | 0.0 | 10.0 | 0.0 | 0.0 | 18.8 | 0.0 | 8.1 |
| Qwen3-VL-30B-A3B | 22.7 | 4.6 | 18.8 | 0.0 | 0.0 | 0.0 | 5.9 | 6.3 | 10.0 | 8.7 |
| Qwen3-VL-8B | 18.2 | 4.6 | 18.8 | 0.0 | 0.0 | 0.0 | 0.0 | 6.3 | 10.0 | 6.7 |
| GUI-Owl-7B | 13.6 | 0.0 | 18.8 | 0.0 | 0.0 | 11.1 | 2.9 | 12.5 | 0.0 | 6.7 |
| MA3(GUI-Owl-7B) | 18.2 | 9.1 | 6.3 | 0.0 | 0.0 | 0.0 | 11.8 | 31.3 | 20.0 | 12.1 |
| Kimi K2.5 | 40.9 | 0.0 | 50.0 | 10.0 | 0.0 | 0.0 | 20.6 | 43.8 | 50.0 | 24.8 |
| *Closed-source* | | | | | | | | | | |
| Gemini-3-Pro + UI-Venus-72B | 54.6 | 4.6 | 56.3 | 20.0 | 0.0 | 11.1 | 41.2 | 75.0 | 40.0 | 36.9 |
| GPT-5.1 + UI-Venus-72B | 54.6 | 0.0 | 31.3 | 0.0 | 0.0 | 11.1 | 26.5 | 56.3 | 40.0 | 26.9 |
| Seed 2.0 | 18.2 | 4.6 | 43.8 | 10.0 | 0.0 | 11.1 | 17.7 | 31.3 | 20.0 | 18.1 |
| **Average** | **27.7** | **3.7** | **28.2** | **4.0** | **2.0** | **4.4** | **14.4** | **33.2** | **19.0** | **16.4** |

**Per-step Tokens (PT)** measures the average output token volume per decision step:

$$PT = \frac{TT}{\sum_{i=1}^{N} T_i}. \tag{5}$$

## 4.2. Holistic Evaluation on VenusBench-Mobile

Table 3 reports the performance of evaluated mobile GUI agents on VenusBench-Mobile. The results reveal several insights that are not exposed by existing benchmarks:

> **Finding 1:** VenusBench-Mobile addresses critical real-world needs ignored by prior benchmarks, thereby uncovering the true performance deficits of current agents and establishing a highly discriminative standard to guide future model advancement.

**Substantial Difficulty Increase.** As shown in Table 3, the success rates on VenusBench-Mobile are significantly lower than those reported on traditional benchmarks. Even the most capable model, Gemini-3-Pro, achieves a total success rate of only 36.9%, while most open-source models struggle below 15%. This stark contrast indicates that our benchmark successfully moves beyond simple instruction following to test the limits of current agents in realistic environments.

**Exposure of Blind Spots in Prior Benchmarks.** VenusBench-Mobile covers a wide range of real-world user needs, particularly GSA and GUIM, which have not appeared in any prior benchmarks summarized in Table 1. As shown in Table 3, these categories also represent the most challenging scenarios for current agents, with average success rates of only **2.0% (GSA)** and **4.4% (GUIM)**. This highlights a critical oversight: by ignoring these prevalent real-world user needs, prior benchmarks failed to uncover

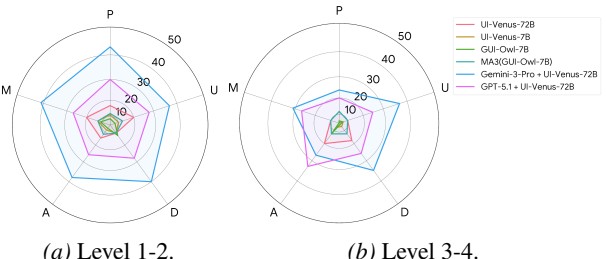

*(a)* Level 1-2.      *(b)* Level 3-4.

*Figure 6.* Agent performance $Acc_{dim}$ (%) across capability levels.

the true performance deficits of recent agents, thereby failing to provide meaningful guidance for model advancement.

**High Discriminative Power.** VenusBench-Mobile effectively differentiates model capabilities. Unlike simpler benchmarks where performance quickly saturates, our benchmark reveals a clear performance hierarchy. For example, on the challenging NR task, Gemini-3-Pro achieves a 75.0% success rate, while smaller models such as Qwen3-VL-8B drop to 6.3%. This large performance gap demonstrates that VenusBench-Mobile can reliably distinguish robust general-purpose agents from weaker models that rely on superficial pattern matching.

## 4.3. Fine-grained Capability Diagnosis via PUDAM

> **Finding 2:** The PUDAM taxonomy validates the hierarchical nature of agent capabilities, shifting evaluation from simple outcomes to root-cause diagnosis. Our analysis reveals that while open-source models suffer a specific collapse in high-level *Decision (D)* and *Perception (P)*, *Memory (M)* remains the absolute bottleneck across the board, preventing the evolution from basic automation to sophisticated embodied intelligence.

*Table 4.* Stability Pass Rate (SPR, %) under different environment perturbations, where success requires completion across all five settings.

| Model | Environment Settings | | | | | Summary | | |
|---|---|---|---|---|---|---|---|---|
| | Original | Chinese | Variation | Dark | Pad | Min | Max | **SPR** |
| UI-Venus-72B | 15 | 15 | 20 | 15 | 5 | 5 | 20 | 0 |
| UI-Venus-7B | 15 | 20 | 15 | 10 | 5 | 5 | 20 | **5** |
| Qwen3-VL-30B-A3B | 10 | 10 | 15 | 10 | 5 | 5 | 15 | 0 |
| Qwen3-VL-8B | 10 | 20 | 15 | 10 | 10 | 10 | 20 | 0 |
| GUI-Owl-7B | 15 | 20 | 10 | 15 | 0 | 0 | 20 | 0 |
| MA3(GUI-Owl-7B) | 10 | 15 | 15 | 20 | 10 | 10 | 20 | 0 |
| Gemini-3-Pro + UI-Venus-72B | 35 | 35 | 25 | 35 | 30 | 25 | 35 | **15** |
| GPT-5.1 + UI-Venus-72B | 40 | 15 | 35 | 30 | 20 | 15 | 40 | **5** |

Overall success rates provide only a coarse snapshot of performance and do not explain the underlying causes of failure. By mapping tasks to the PUDAM taxonomy (detailed in Section A), analyzing capability-level performance in Figure 6 and detailed information in Table 8, we obtain deeper insights into agent behavior.

**Validation of Taxonomy Rationality.** The empirical results in Table 8 strongly validate the hierarchical design of our PUDAM taxonomy. By contrasting the paired columns, we observe a consistent performance drop from Basic (L1-2) to Advanced (L3-4) levels across essentially all dimensions (e.g., the average Perception score drops from 17.5% to 10.3%). This trend confirms that our level definitions and data annotations accurately reflect increasing task complexity, proving that the taxonomy functions as a reliable, high-quality standard for distinguishing between fundamental automation and sophisticated intelligence.

**Decision & Perception Collapse under Complexity.** While open-source models can follow deterministic paths at lower levels, their capabilities falter significantly at higher levels where *Decision (D)* and *Perception (P)* become critical. As shown in Table 8, the success rates of most open-source models drop to single digits ($<$5%) in L3-4 Decision and Perception. This collapse highlights a fundamental deficiency: current smaller models lack the *autonomous reflection* needed to correct errors and the *dynamic temporal perception* required to handle environmental noise, limiting them to rigid, script-like execution.

**Memory as the Absolute Bottleneck.** The analysis identifies *Memory (M)* as the most severe bottleneck for agent evolution. As evidenced in Table 8, performance in *Memory* suffers the steepest catastrophic drop at advanced levels. While proprietary models maintain some functionality, smaller open-source models like UI-Venus-7B and GUI-Owl-7B plummet to near-zero success rates in L3-4. This indicates a systemic inability to aggregate cross-screen information or maintain context in long-sequence tasks, suggesting that simply increasing context window size is insufficient without dedicated state-tracking mechanisms.

### 4.4. Stability Analysis

> **Finding 3:** Systematic environment variations in VenusBench-Mobile reveal that current agents are highly brittle, with near-zero Stability Pass Rates showing poor generalization. Visual layout changes (e.g., Tablet mode) lead to the largest performance drops across all models, while language variations expose inconsistent robustness even in the strongest proprietary agents.

We evaluate agent consistency under four systematic perturbations: Chinese translation, instruction variation, dark theme, and tablet (Pad) orientation. The results in Table 4 lead to three critical observations:

**Extreme Brittleness Revealed by SPR.** The Stability Pass Rate (SPR), which requires an agent to succeed across all five variants of the same task, reveals a clear reliability issue uncovered by our benchmark. Most models achieve an SPR of 0%, and even the best-performing model reaches only 15%. This shows that high success on standard settings does not translate to stable behavior. Our benchmark demonstrates that small, realistic changes are often enough to break current agents.

**High Sensitivity to Visual Layouts.** Controlled layout perturbations show that Tablet (Pad) mode causes the largest performance drop. Many open-source agents fall to near-zero success rates, exposing a strong dependence on vertical phone layouts. This highlights a key limitation: current agents struggle to adapt when UI structures change, even though the underlying task remains the same.

**Unstable Linguistic Robustness.** By varying language while keeping tasks and interfaces fixed, our benchmark reveals large differences in linguistic robustness. Some models remain stable across languages, while others suffer sharp drops in performance. This suggests that multilingual GUI understanding is still fragile, and our benchmark provides a systematic way to diagnose this weakness beyond overall success rates.

*Table 5.* Comparison of inference efficiency through Total Tokens and Per-Step Tokens.

| Model | Total | Per-Step |
|---|---|---|
| UI-Venus-72B | 850.0K | 153.2 |
| UI-Venus-7B | 447.4K | 101.5 |
| Qwen3-VL-30B-A3B | 463.2K | 138.9 |
| Qwen3-VL-8B | 357.7K | 132.4 |
| GUI-Owl-7B | 373.7K | 146.5 |
| MA3(GUI-Owl-7B) | 1640.0K | 438.7 |
| Gemini-3-Pro + UI-Venus-72B | 259.7K | 86.3 |
| GPT-5.1 + UI-Venus-72B | 167.5K | 54.6 |

### 4.5. Inference Cost

We prioritize token counts over execution time because the latter is highly dependent on specific hardware configurations and network environments, whereas tokens provide a standardized measure. Furthermore, token consumption directly reflects the actual economic costs and API billing for both open-source and proprietary models. Therefore, we examine computational cost through Total Tokens and Per-Step Tokens in Table 5.

**Model Scale and Efficiency.** There is a clear correlation between model scale and token consumption among open-source agents. UI-Venus-72B consumed 850.0K total tokens, nearly double that of the UI-Venus-7B at 447.4K. In the proprietary category, GPT-5.1 proves to be the most efficient, with the lowest total (167.5K) and per-step (54.6) token usage, suggesting highly optimized visual encoding or input compression.

**Agent Framework Overhead.** The most significant finding is the massive overhead introduced by agentic reasoning loops. MA3 (based on GUI-Owl-7B) consumed a staggering 1.64M total tokens, with a per-step cost of 438.7. This represents roughly a $3.0\times$ increase compared to the standalone GUI-Owl-7B base model (146.5 tokens per step). While such frameworks aim to improve success through reflection and planning, the resulting token density poses a critical barrier for real-time deployment on edge devices where battery life and bandwidth are limited.

### 5. Conclusion

In this work, we presented VenusBench-Mobile, a challenging online benchmark for evaluating mobile GUI agents under realistic, user-centric conditions. By adopting a user-intent–driven task design and a capability-oriented diagnostic annotation scheme, VenusBench-Mobile enables evaluation beyond app-centric success rates and supports fine-grained analysis of agent behaviors. Extensive experiments show that state-of-the-art agents exhibit large performance drops relative to prior benchmarks, revealing fundamental limitations in perception, memory, and robustness. These findings suggest that current mobile GUI agents remain far from reliable real-world deployment. We believe VenusBench-Mobile provides a foundational evaluation benchmark to guide future research toward more robust and general-purpose mobile GUI agents.

### Acknowledgements

This work was supported by the Fundamental and Interdisciplinary Disciplines Breakthrough Plan of the Ministry of Education of China under Project No. JYB2025XDXM114. This work was also supported by the Ant Group Research Intern Program.

### Impact Statement

This paper presents work whose goal is to advance the field of machine learning. There are many potential societal consequences of our work, none of which we feel must be specifically highlighted here.

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

# A. Four Proficiency Levels Across Five Dimensions

We define four proficiency levels across five capability dimensions to systematically characterize GUI agent competencies. Each dimension progresses hierarchically from basic functionality to general embodied intelligence, with some dimensions not applicable at lower levels due to their inherently advanced nature. To concretely illustrate what each proficiency level entails in practice, we provide representative task examples for each level within every dimension.

### A.1. Perception (P): Understanding GUI Environment

**P1 - Static Visual Element Recognition**: Recognizes text, standard icons, and UI elements through OCR and basic visual processing. Capable of identifying individual controls and their labels in static interfaces.

**Example Task:** *Explore pro expense and tell me how to add a new expense. Specifically, identify all available fields, options, and settings that can be configured when creating an expense.*

**Explanation:** The agent only needs to recognize static labels and input fields in the Pro Expense creation form to list available configuration options.

**P2 - Semantic Layout Understanding**: Comprehends overall layout structure and multi-modal associations between text and images. Understands the hierarchical relationships between UI components.

**Example Task:** *There is a picture named graphic.png. Use the Gallery app to open the image. Adjust the picture so that the green rectangle is at the top and the red circle is on the right side. After setting, do not save the picture. Just leave the canvas on-screen so the user can see the result.*

**Explanation:** The agent must comprehend the spatial layout and semantic properties (color and shape) of elements within the canvas, requiring an understanding of the overall visual structure to perform requested rearrangements.

**P3 - High-Precision Localization**: Achieves pixel-level precise localization of tiny or implicit visual elements.

**Example Task:** *Among the apps Markor, Joplin, Tasks, Broccoli, Simple Calendar Pro, VLC, and Audio Recorder, identify the one whose settings screen contains four exclamation-mark buttons, each used to explain the meaning of the setting it is next to. Provide the app name.*

**Explanation:** Success depends on the pixel-level localization of tiny, symbolic icons (exclamation marks) and correctly associating them with adjacent setting text.

**P4 - Dynamic Temporal Perception**: Perceives dynamic temporal changes including animations, video streams, key frames, and UI states evolving over time. Capable of tracking and interpreting interface elements that change continuously.

**Example Task:** *Record an audio clip in Audio Recorder, pausing at 14-16 seconds (do not save).*

**Explanation:** This task demands real-time monitoring of a dynamic timer and precise action triggering based on the evolving UI state over a specific time window.

### A.2. Understanding (U): Comprehending User Instructions

**U1 - Atomic Instruction Understanding**: Understands single operation intents with simple action-object pairs (e.g., "click the confirm button").

*VenusBench-Mobile has no test cases for U1 level.*

**U2 - Deterministic Task Instruction Understanding**: Interprets explicit task instructions and their combinations (e.g., "open Markor, create a note, record 'abc'").

**Example Task:** *Record an audio clip and save it with name discussion_copy.m4a using Audio Recorder app.*

**Explanation:** The instruction provides an explicit sequence of steps with deterministic parameters (record an audio clip, save it), allowing for a direct execution path without ambiguity.

**U3 - Complex Instruction Understanding**: Comprehends explicit task instructions combined with sophisticated constraint conditions (e.g., "Find apps that support 50 languages AND theme changes."). The key challenge lies in parsing instructions with multiple constraints, such as multi-dimensional evaluation criteria with logical operators, implicit comparisons, and

reasoning-based requirements.

**Example Task:** *Among Audio Recorder, Draw, Vlc, Pro Expense, Broccoli, MediLog and Markor, some apps support changing the interface language and the theme color. Among the apps that meet the requirements, find the one that offers more than 50 supported languages in its in-app settings. Tell me the name of the APP.*

**Explanation:** The agent must navigate multiple layers of constraints, including app lists and quantitative thresholds, requiring multi-step logical reasoning to fulfill the request.

**U4 - Ambiguous/Vague Instruction Understanding**: Interprets ambiguous or information-insufficient instructions. May need to combine GUI environment context and historical interaction patterns. The difficulty lies in identifying ambiguity, recognizing instruction incompleteness, and disambiguating through reasoning with GUI environment.

**Example Task:** *Change the Color Theme in Tomoto app to black.*

**Explanation:** The instruction appears straightforward but contains implicit ambiguity—the requested "black" theme does not exist in Tomoto's available color options. The agent must recognize this environmental constraint mismatch and reason that the instruction is unachievable, requiring clarification from the user rather than attempting impossible execution.

### A.3. Decision (D): Strategic Reasoning During Execution

**D1**: Not applicable—basic-level agents may follow predetermined scripts without decision-making capabilities.

*VenusBench-Mobile has no test cases for D1 level.*

**D2 - Deterministic Execution**: Executes tasks along relatively clear and predetermined paths. The agent follows straightforward plans where the sequence of actions is largely specified or obvious from the task structure.

**Example Task:** *Navigate to the setting in Tasks APP that allows you to change the task sort order and show me that interface.*

**Explanation:** The agent follows a straightforward and predetermined navigation path within the Tasks APP to locate a specific setting, following a sequence of obvious steps to reach the target UI.

**D3 - Dynamic Strategy Adaptation**: Employs dynamic decision-making and real-time task path adjustment. Handles environmental noise (popups, ads, unexpected dialogs) and makes context-aware selections based on GUI information and trajectory history. Capable of dynamic reasoning and filtering (e.g., comparing prices across multiple options and selecting based on criteria).

**Example Task:** *There are some screenshot located at the folder GUIBrowsing/figure1. In the current screenshot of the Meituan food-delivery app (open it in Gallery), help me find the cheapest Americano that can be delivered within 30 minutes. Return the answer in Chinese in the following format: merchant full name, the item full name.*

**Explanation:** The agent must engage in dynamic reasoning and filtering, comparing real-time price and delivery data across various options to make an optimal selection based on multiple criteria (price and delivery time).

**D4 - Reflective Planning**: Demonstrates enhanced autonomy through self-reflection capabilities. Handles conflicts, explores alternative paths when primary routes fail, performs error correction, intercepts high-risk operations, and improves robustness through cautious verification strategies.

**Example Task:** *Open the Camera app, switch to Video mode, set the exposure compensation to –1 EV, then start recording and capture a 10-20 second clip.*

**Explanation:** This level requires the agent to demonstrate robustness by managing potential hardware setting conflicts (one cannot set the exposure compensation to –1 EV when the Camera app is switched to Video mode) and reflecting on the device state before initiating operations, potentially reordering steps or requesting user clarification.

### A.4. Action (A): Physical GUI Interaction

**A1 - Basic Operations**: Performs fundamental touch interactions including tap and long press on clearly defined interactive elements.

*VenusBench-Mobile has no test cases for A1 level.*

**A2 - Complex Trajectory Operations**: Executes operations requiring controlled movement including linear/non-linear scrolling and dragging across various distances and directions.

**Example Task:** *Find a phone screen that lets me toggle Bluetooth and Airplane mode, as well as adjust the brightness, all in one place. Navigate to that screen and show it to me.*

**Explanation:** This task involves navigating through system settings, which typically requires a series of precise, multi-directional swipes and scrolling actions to reach the specific consolidated control panel (quick settings panel).

**A3 - Precision Operations**: Performs precision operations requiring semantic understanding and spatial accuracy, such as cursor positioning at specific text locations, fine slider adjustment to target values, and navigation through nested menu hierarchies.

**Example Task:** *The screen is too dark. I can see nothing.*

**Explanation:** Success depends on high spatial accuracy to manipulate a fine brightness slider control, requiring the agent to first locate the brightness adjustment interface and then perform precise dragging to achieve the desired brightness level.

**A4 - Real-Time Closed-Loop Control**: Achieves millisecond-level hand-eye coordination with continuous visual feedback integration. Capable of real-time trajectory correction during execution (e.g., dynamic drawing with ongoing path adjustment based on visual feedback).

*VenusBench-Mobile has no test cases for A4 level.*

**A.5. Memory (M): Task-Relevant Information Memorization**

**M1**: Not applicable. Basic-level agents may operate statelessly without explicit memory mechanisms.

*VenusBench-Mobile has no test cases for M1 level.*

**M2 - Task Path Maintenance**: Maintains task objectives and trajectory history across application and page transitions without losing the overall goal. Ensures continuity of purpose throughout multi-step workflows spanning different interface contexts.

**Example Task:** *Launch the Tomato app, let the long-break timer run for about 19-21 seconds, then hit Pause.*

**Explanation:** The agent must maintain the specific task goal (pausing at a certain time) throughout the workflow without losing focus on the temporal objective, even as the interface displays dynamic countdown information.

**M3 - Long-Term State Tracking**: Tracks extended state evolution by maintaining both trajectory history and dynamic information pools. Extracts and preserves critical task-relevant information from historical interactions (e.g., aggregating product prices and features across multiple pages for comparison-based decision making).

**Example Task:** *Handle the following 8 SMS. Every time a new SMS arrives, read its body and check if it contains any of the keywords ['urgent', 'meeting', 'password', 'alert'] (case-insensitive). Record the content of the messages with keywords in message.md in Markor. Format each line as 'Contact Name: Message' and record them in the order they arrive.*

**Explanation:** The agent must extract and aggregate critical information from a series of independent external triggers (SMS messages) into a persistent, dynamic information pool, maintaining both filtering logic and ordering constraints across multiple discrete events.

**M4 - Long-Term Cross-task Memory**: Retains and retrieves information across multiple completed tasks. Maintains a persistent memory of entities, actions, and temporal ordering from previous task executions, enabling retrospective operations that require recalling specific details from earlier tasks (e.g., "delete the first two notes created earlier").

**Example Task:**

*Round 1 - Create a new note in Markor named 2023_05_25_neat_frog.md with the following text: May the Force be with you.*

*Round 2 - Create another note named g40M_helpful_mouse.md with the following text: Your dentist appointment is scheduled for 2 PM on Thursday.*

*Round 3 - Create another note named good_igloo_copy.md with the following text: I think, therefore I am.*

*Round 4 - Create another note named 2023_02_22_good_nest.md with the following text: The dog's vet appointment is next Monday at 11 AM.*

*Round 5 - Delete the first two notes.*

**Explanation:** This task requires the agent to recall information from previous task sessions (Rounds 1-4) to correctly identify and delete "the first two notes" in Round 5. The agent must maintain a persistent memory of file names and creation order across completed tasks, demonstrating cross-session experience accumulation.

## B. Benchmark Setting Details

In this section, we provide additional details of our benchmark setting, including task curation, action space, app list, setting of MLLM-as-a-Judge.

### B.1. Details of Primary Task Curation

**Primary Task Pool.** Based on our task taxonomy and selected applications, we manually construct tasks for nine categories (FA, CF, VA, MR, GSA, GUIM, HGB, NR, BC), resulting in a primary pool of 149 tasks. Figure 3 shows the distribution across categories. All tasks are designed to run on Android emulators with standard phone resolution (1080×2400) in light mode.

**Data Quality Assurance.** We perform two-stage quality assurance: manual auditing of all tasks (queries, initialization, and verification) and PUDAM annotations cross-validated by independent reviewers. As shown in Figure 4, AndroidWorld is concentrated at lower capability levels with very limited Level-4 coverage, while VenusBench-Mobile provides broader coverage across Levels 2–4, enabling evaluation of advanced capabilities.

### B.2. App List

Table 6 presents the complete list of 27 applications used in VenusBench-Mobile, including the 20 apps from AndroidWorld and 7 newly added apps to enhance GUI diversity and coverage of real-world usage scenarios. We also present the task distribution across apps. "# Tasks (Primary + Subset)" indicates the number of tasks from the primary pool (149 tasks) and additional 80 variant tasks from the Stability Evaluation subset, respectively.

### B.3. Action Space

Our benchmark supports a set of actions that cover common operations for GUI agents, as detailed in Table 7.

### B.4. MLLM-as-a-Judge Setting

For the 90 tasks (out of 149) whose verification requires visual recognition or semantic understanding (e.g., all tasks in the FA category), we employ Qwen3-VL-30B-A3B-Instruct with carefully crafted prompts for MLLM-based evaluation. Below, we detail the 7 verification interfaces employed in our framework, with representative test cases to demonstrate each interface type.

#### B.4.1. JUDGE 1: FACT MATCHING FOR BC AND HGB TASKS

**Task Description & Evaluation Logic.** This interface is designed for BC and HGB tasks that possess relatively fixed and deterministic ground-truth answers. The primary challenge in evaluating these tasks lies in the variability of model outputs; for instance, if the ground truth is "Pro Expense," a model might respond with "Expense" or "The answer is Pro Expense", both of which should be considered correct. However, if a model outputs multiple potential options or irrelevant content (e.g., "Pro Expense and Markor"), it must be judged as incorrect. The evaluation logic focuses on factual equivalence while strictly prohibiting over-generation.

**Prompt.** To implement the above evaluation logic, we design a specialized prompt that instructs the MLLM judge to perform strict fact matching. The complete prompt structure is illustrated below, consisting of both a system-level instruction and a user-level query template.

---
**Fact Check Prompt**

```
System Prompt:
You are a mobile GUI task verifier.
Decide whether the predicted textual answer expresses the SAME FACT as the
ground-truth answer.
Ignore capitalization, punctuation, white-space and minor wording differences.
The prediction cannot be empty; otherwise it will be immediately judged INCORRECT.
Two answers are considered the SAME when:

  1. All key factual values (numbers, dates, names, file names, paths, etc.)  are
     identical.

  2. The core assertion described is factually equivalent|even if phrased
     differently.

  3. No content beyond the ground truth answer is allowed; otherwise, it will be
     considered incorrect.

User Template:
TASK GOAL: {goal}
PREDICTED ANSWER: {prediction}
GROUND-TRUTH ANSWER: {gt_answer}
Is the predicted answer match the ground-truth answer?
```
---

**Illustrative Examples.**  To demonstrate the behavior of this evaluation interface in practice, we present a representative test case with both positive and negative examples.

- **Task Goal (Query):** Among Files, Joplin, Tasks, Broccoli, Pro Expense, Markor and Calendar, open an app whose icon contains blue color. After launching, the UI should not have any search bar or search button at the top. Find the APP and tell me its name.

- **Ground-truth Answer:** Pro Expense

- **Positive Example:** "Expense" (Result: ✓ **Correct** — Focuses on the core factual name).

- **Negative Example:** "Pro Expense, Markor and Calendar" (Result: ✗ **Incorrect** — Contains content beyond the ground truth).

B.4.2. JUDGE 2: SEMANTIC VERIFICATION FOR FA TASKS

**Task Description & Evaluation Logic.**  This interface is designed for FA tasks, where the agent is required to introduce an app's features or provide a step-by-step tutorial to achieve a specific goal. While the target objectives are deterministic, the natural language descriptions can be diverse and detailed. The evaluation logic requires the agent to cover the majority of key steps or core functionalities listed in the ground truth to ensure the user can successfully complete the task. While minor omissions or phrasing variations are permitted, the prediction must not introduce hallucinated features or contradictions that could lead to user misunderstanding.

**Prompt.**  To implement the semantic verification logic described above, we design a prompt that guides the LLM judge to assess coverage and factual consistency rather than exact textual matching. The complete prompt specification is presented below.

---
**Function Assistance Prompt**

```
System Prompt:
You are a mobile GUI task verifier for FuncAssist missions.
Decide whether the predicted textual answer semantically affirms most of the facts
expressed in the ground-truth answer.
Treat the ground truth as a set of facts.
```
---

```
Accept paraphrases, synonyms, and logically equivalent formulations.  Ignore
capitalization, punctuation, and minor formatting.

Crucial Criteria:

 1. For functionality introductions, mark INCORRECT if the prediction covers less
    than half of the ground-truth functionalities or includes any features not
    present in the ground truth.

 2. Answer ''yes'' only if the prediction supports most ground-truth facts and does
    not introduce contradictions.

 3. Answer ''no'' if it misses critical facts that cause misunderstanding or asserts
    details that conflict with the ground truth.

User Template:
TASK GOAL: {goal}
PREDICTED FUNCTIONALITIES: {prediction}
GROUND-TRUTH FUNCTIONALITIES: {gt_answer}
Does the predicted list accurately cover all functionalities listed in the
ground-truth?
```

**Illustrative Examples.**    To demonstrate the behavior of this evaluation interface in practice, we present a representative test case with both positive and negative examples.

- **Task Goal (Query):** Please tell me how to use ZipXtract to extract files.

- **Ground-truth Answer:** First, click the button below and select 'Extract'. Then click 'Pick File' to choose the folder path. Next, click 'Extract to' to select the destination path. Finally, click 'Extract' to complete the extraction.

- **Positive Example:** "Open ZipXtract and begin by clicking the button at the bottom, then select 'Extract'. After that, tap 'Pick File' to choose the folder path... Finally, click 'Extract' to complete the extraction." (Result: ✓ **Correct** — Semantically equivalent and covers all key steps).

- **Negative Example:** "To use ZipXtract, just open the app and it will automatically extract all compressed files on your device without needing you to select any file..." (Result: ✗ **Incorrect** — Introduces contradictory logic and misses essential manual steps).

### B.4.3. JUDGE 3: CONFLICT RESOLUTION AND CF TASKS

**1. Task Description & Evaluation Logic:**
This interface is designed for CF tasks, where agents must identify and resolve contradictions or ambiguities between user instructions and the actual GUI state. In such scenarios, the agent is expected to recognize that the task is unachievable and perform a "Call User" action to prompt for respecification. The evaluation logic focuses on whether the agent correctly refuses the impossible request and provides a semantically accurate reason for the refusal. Any claim of successful execution for a conflicting task is judged as a critical failure.

**Prompt.**    To evaluate the agent's refusal behavior and reasoning quality, we design a prompt that instructs the LLM judge to verify both the refusal decision and the semantic accuracy of the explanation. The prompt structure is shown below:

**MLLM Conflict Verifier Prompt**

```
System Prompt:
You are a mobile GUI task verifier for Refusal missions.
Your task is to decide whether the predicted textual answer is the same as the
ground-truth answer.
Ignore capitalization, punctuation, and spacing.
Focus on values such as numbers, dates, and names.
```

```
Crucial Criteria:

 1. Judge whether the PREDICTION semantically affirms most of the facts expressed in
    the GROUND-TRUTH answer.

 2. The prediction cannot be empty.

 3. Answer ``no'' if the prediction misses important ground-truth facts which might
    cause misunderstanding, contradicts the ground truth, or asserts successful
    execution of a task that should be refused.
```

**User Template:**
```
TASK GOAL: {goal}
PREDICTED ANSWER: {prediction}
GROUND-TRUTH ANSWER: {gt_answer}
Is the predicted answer match the ground-truth answer?
```

**Illustrative Examples.** To illustrate how this interface evaluates conflict detection and refusal reasoning, we present a representative case where the user requests an unavailable feature.

- **Task Goal (Query):** Change the Color Theme in Tomoto app to black.

- **Ground-truth Answer:** Sorry, I can't set the color theme to black; that color isn't available in the Color Theme of Tomoto app.

- **Positive Example:** "I can't change the color theme to black because it isn't provided as an option in Tomoto's Color Theme settings." (Result: ✓ **Correct** — Accurately identifies the conflict and explains the reason).

- **Negative Example:** "I've successfully changed the Color Theme in Tomoto app to black." (Result: ✗ **Incorrect** — Fails to detect the environmental conflict and hallucinates success).

### B.4.4. JUDGE 4: VISUAL PAGE LOCATING TASKS

**Task Description & Evaluation Logic.** This interface is designed for Locating tasks, where the agent is required to navigate the user to a specific GUI page. These target pages typically have a unique and fixed layout. Evaluation is performed by an MLLM-as-a-Judge by comparing the agent's final screenshot with a ground-truth reference image. The evaluation focuses on the core visual structure, visible widgets, and functional layout. Minor dynamic elements such as system time-stamps, battery levels, or network icons are ignored, as they do not affect the identity of the functional interface.

**Prompt.** To enable robust visual verification, we design a prompt that instructs the MLLM judge to compare screenshots based on structural equivalence rather than pixel-perfect matching. The complete prompt specification is presented below:

**Locating Verifier Prompt**

**System Prompt:**
```
You are a GUI interface-locating verifier.
Your job is to decide whether the last mobile screenshot is the same interface/page
as the ground-truth reference screenshot.
Ignore minor visual differences such as time-stamps, battery level, or temporary
pop-ups.  Focus on the core layout, visible widgets, and overall visual structure.
```

**User Template:**
```
TASK GOAL: {goal}
=== GROUND-TRUTH REFERENCE IMAGE ===
{gt_image_placeholder}
=== LAST MOBILE SCREENSHOT ===
{last_image_placeholder}
Is the last mobile screenshot showing the same interface as the ground-truth
reference?
```

**Illustrative Examples.** To clarify how visual verification operates in distinguishing correct from incorrect page navigation, we present test cases below. The ground truth screenshot, positive example and negative example are presented in Figure 7.

- **Task Goal (Query):** Explore Pro Expense and show me the interface where I can add a new expense.

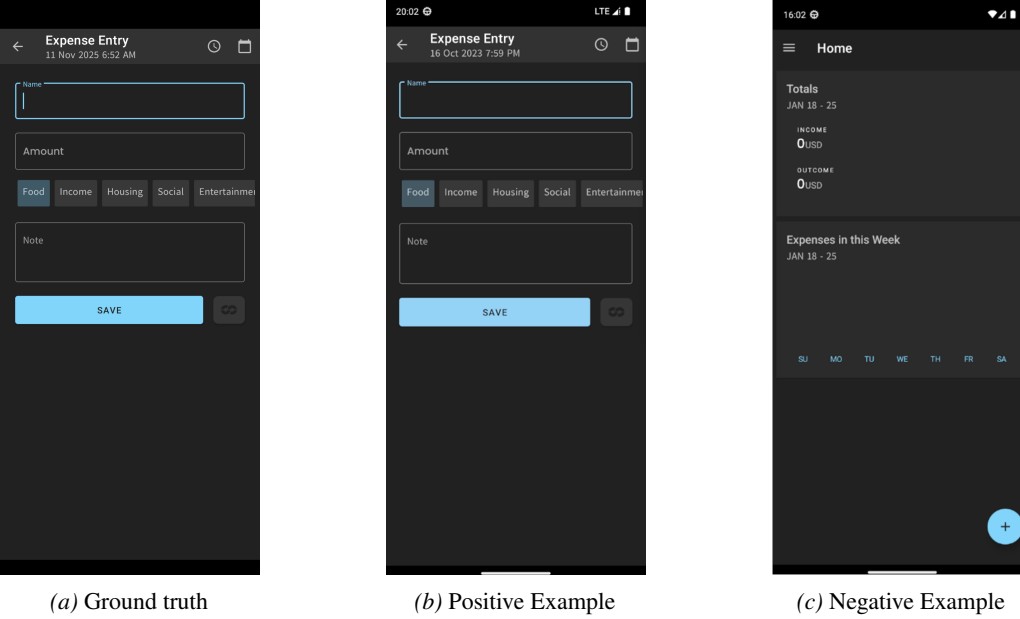

| *(a)* Ground truth | *(b)* Positive Example | *(c)* Negative Example |

*Figure 7.* Verification for Locating APP Interface Tasks. **Positive Case**: The core layout is identical despite a different system time, indicating correct localization. **Negative Case**: Although the correct app is open, the layout differs from the ground truth, indicating the agent is on the wrong page.

### B.4.5. JUDGE 5 VIDEO CONTENT VERIFICATION

**Task Description & Evaluation Logic.** This interface is designed for Video-related tasks, such as finding a video that matches a specific description or pausing a video at a precise moment. Due to the temporal continuity and high frame rate of videos, a task goal may be satisfied by multiple different frames, making a single fixed ground-truth image inappropriate. Instead, this interface employs an MLLM-as-a-Judge to directly assess the final screenshot against the user's query. The judge determines whether the visual content in the screenshot fulfills the intent and whether the key elements (e.g., specific objects, actions, or themes) are correctly represented.

**Prompt.** To enable direct goal-based visual assessment, we design a prompt that instructs the MLLM judge to evaluate task completion by interpreting both the user's intent and the screenshot content. The prompt structure is shown below:

---

**MLLM Video Task Verifier Prompt**

```
System Prompt:
You are a mobile GUI task verifier.
Your task is to decide whether task has been COMPLETED based on:

 1. The task goal description,

 2. The final screenshot.  Carefully examine the screenshot to check if the content
    matches the goal requirements.

 3. Consider whether the main intent is fulfilled and whether key elements mentioned
    in the goal are present.

User Template:
VIDEO TASK GOAL: {goal}
= FINAL SCREENSHOT =
```

---

```
{image_placeholder}
Based on the screenshot above, has the task been completed according to the goal?
```

**Illustrative Examples.** To illustrate how this goal-based verification distinguishes successful from unsuccessful video task completion, we present a representative case below. The positive example and negative example are shown in Figure 8:

- **Task Goal (Query):** I want to watch a local video about water sports. Do not exit after completing the task.

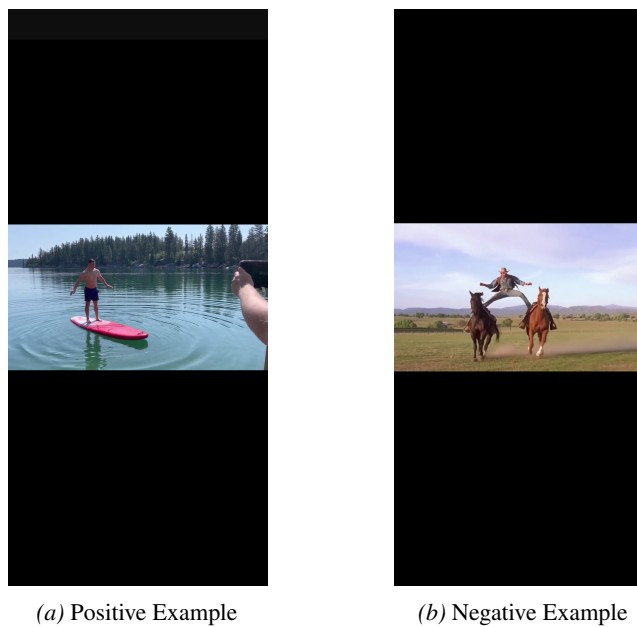

*(a)* Positive Example          *(b)* Negative Example

*Figure 8.* Verification for Watching a Local Video Tasks. **Positive Case**: The screenshot displays an active video player showing water sports, fulfilling the goal. **Negative Case**: The screenshot shows a video listing or unrelated content, failing the requirement.

### B.4.6. JUDGE 6: GUIM-DRAWING VERIFICATION

**1. Task Description & Evaluation Logic:**
This interface is tailored for drawing tasks within the GUIM category. Unlike functional automation, drawing tasks yield highly diverse visual outputs with no unique standard answer. Consequently, this interface does not utilize a static ground-truth image. Instead, an MLLM-as-a-Judge evaluates the final canvas against the task description. The verification focuses on geometric fidelity (e.g., shapes, relative positions), color accuracy, content representation, and quantitative requirements (e.g., the number of items). The judge must also ensure the drawing is reasonably complete (e.g., all sides of a polygon are rendered).

**Prompt.** To enable comprehensive geometric and compositional assessment, we design a prompt that guides the MLLM judge through multiple verification dimensions—shape correctness, color compliance, and structural completeness. The prompt specification is presented below:

---
**MLLM Drawing Verifier Prompt**

```
System Prompt:
You are a mobile GUI drawing task verifier.
Your task is to decide whether the drawing task has been COMPLETED based on:
(1) the task goal description (what should be drawn), and (2) the final canvas
screenshot.
Carefully examine the screenshot to check if the drawn content matches the goal
requirements.
```

```
Consider the following aspects:

  • Shape & Content:  Does the drawing have the correct basic shapes (circle,
    rectangle, etc.)  and depict the requested objects?

  • Color & Detail:  Does it use the required colors and meet specific details (e.g.,
    uppercase/lowercase letters)?

  • Completeness & Quantity:  Is the drawing complete and does the number of items
    match the query?

User Template:
DRAWING TASK GOAL: {goal}
=== FINAL CANVAS SCREENSHOT ===
{image_placeholder}
Based on the canvas screenshot above, has the drawing task been completed according
to the goal?
```

**Illustrative Examples.** To demonstrate how this interface evaluates geometric constraint satisfaction in drawing tasks, we present a case requiring precise spatial relationships, with visual evidence in Figure 9:

- **Task Goal (Query):** In Joplin app, first draw a circle, then draw a rectangle inside it so that all four vertices of the rectangle lie exactly on the circle's circumference. After drawing, leave the canvas on-screen.

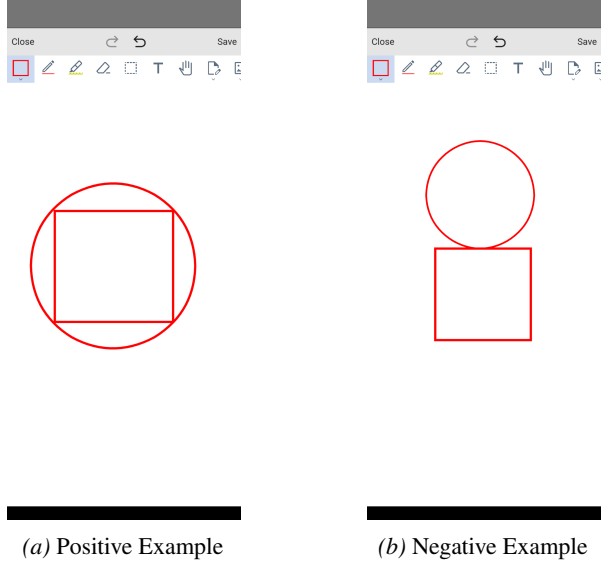

*(a)* Positive Example      *(b)* Negative Example

*Figure 9.* Verification for Drawing Tasks. **Positive Case**: The screenshot shows a circle with a rectangle inscribed correctly according to the geometric constraints. **Negative Case**: The screenshot shows incorrect spatial relationships, failing the goal.

### B.4.7. JUDGE 7: GUIM-EDITING VERIFICATION

**Task Description & Evaluation Logic.** This interface is specialized for Image Editing tasks within the GUIM category, involving operations such as rotation, annotation, and erasing. Similar to drawing tasks, editing results are visually diverse and lack a unique ground truth. The evaluation requires an MLLM-as-a-Judge to verify if the specific edits requested in the query have been accurately executed on the target image. The judge follows strict criteria: for erasing, the target area must be rendered pure white without residual fragments of the original object; for circling, the target must be clearly enclosed without omitting required elements or including extraneous ones.

**Prompt.** To enable operation-specific verification of image transformations, we design a prompt that provides explicit evaluation criteria for common editing operations while maintaining flexibility for diverse task types. The prompt structure

is shown below:

---

**MLLM Image Editing Verifier Prompt**

**System Prompt:**
You are a mobile GUI editing task verifier.
Your task is to decide whether the editing task has been COMPLETED based on: (1)
the task goal description, and (2) the final canvas screenshot.
Carefully examine the screenshot to check if the content matches the goal
requirements.

Consider the following aspects:

  • Erasing: The erased area must be pure white. If any part of the target object
    remains, judge as INCORRECT.

  • Circling: A clearly visible circle must enclose the target. Circling extra
    objects or missing required ones are treated as errors.

**User Template:**
EDITING TASK GOAL: {goal}
=== FINAL CANVAS SCREENSHOT ===
{image_placeholder}
Based on the canvas screenshot above, has the editing task been completed according
to the goal?

---

**Illustrative Examples.** To illustrate the verification process, we present an image editing task where the judge must assess whether the agent correctly identified and marked the target object. Figure 10 shows three scenarios: the original image, a successful completion where only the banana is circled in red, and a failed attempt where all fruits are incorrectly marked.

- **Task Goal (Query):** "There is a picture named fruit.png. Use the Draw app to open the image, circle the banana with a red pen, and stay on the screen after the task is completed."

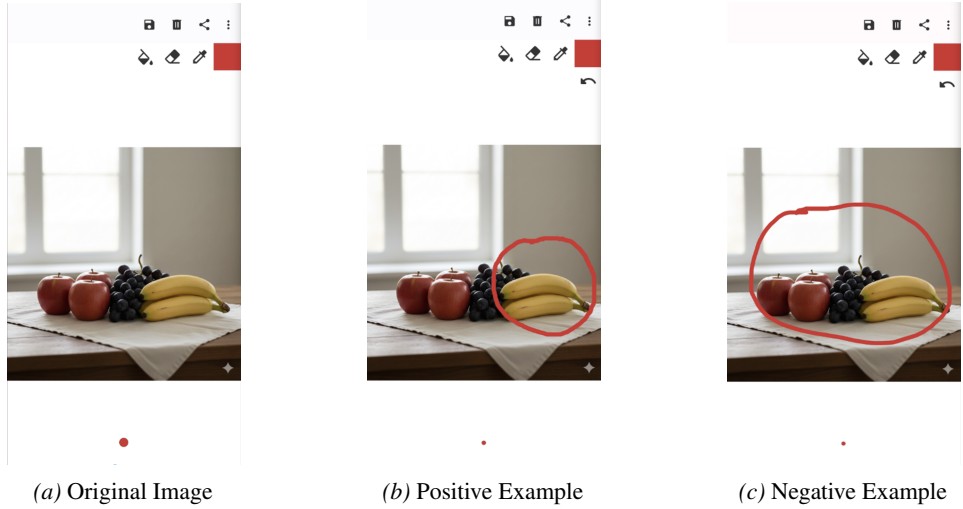

*(a)* Original Image      *(b)* Positive Example      *(c)* Negative Example

*Figure 10.* Verification for Image Editing Tasks. **Positive Case**: Accurately identify the banana and circle it with a red pen. **Negative Case**: Incorrectly circled all the fruits.

## C. Detailed Information of Fine-grained Capabilities

Table 8 presents the detailed numerical breakdown of $Acc_{dim}$ (%) across the five PUDAM dimensions at basic (L1-2) and advanced (L3-4) proficiency levels, corresponding to the radar chart visualizations in Figure 6 of the main text.

The results confirm the key findings discussed in Section 4: (1) **Perception** and **Memory** show the most severe degradation from basic to advanced levels, with average drops of 7.2 and 7.5 percentage points respectively; (2) Open-source models experience catastrophic collapse at L3-4, particularly in Decision and Memory dimensions where most models fall below 10%; (3) Even the best-performing Gemini-3-Pro shows substantial drops in Perception (43% → 24%) and Memory (41% → 31%), validating that these represent fundamental bottlenecks rather than model-scale limitations; (4) MA3(GUI-Owl-7B) demonstrates consistent performance across difficulty levels (10–13%), illustrating how specialized frameworks can stabilize agent behavior despite lower absolute scores.

These detailed numbers provide the empirical foundation for our capability taxonomy validation and bottleneck diagnosis presented in the main paper.

### C.1. Implications

The detailed capability breakdown in Table 8 provides actionable insights for targeted model improvement: (1) **Memory mechanisms** require fundamental architectural innovations beyond simply increasing context windows, as evidenced by the catastrophic L3-4 drops even in large models; (2) **Perception enhancement** should focus on fine-grained spatial reasoning and dynamic state tracking, particularly for handling complex visual layouts and multi-screen information flow; (3) **Decision robustness** remains underdeveloped in open-source models, requiring better error recovery and adaptive planning capabilities.

## D. Agent Framework

To evaluate closed-source models, we construct an agentic framework where both the planner model and summary model are instantiated with the closed-source VLM under evaluation, while a dedicated grounding model is used for precise UI element localization. Our agent operates through a three-stage pipeline executed at each step. **Stage 1: Action Selection.** The agent receives the current screenshot along with the action history (i.e., summaries of previous steps), and queries the planner VLM using the *Action Selection Prompt* to select an action from a predefined action space, which includes clicks, text input, scrolling, and app navigation operations. **Stage 2: Visual Grounding.** For actions requiring target localization (e.g., `click`, `input_text`, `long_press`), the agent employs the visual grounding model with the *Visual Grounding Prompt* to convert the natural language action description from Stage 1 into precise (x, y) screen coordinates. **Stage 3: Execution and Summarization.** The agent executes the selected action. Subsequently, it captures the post-action screenshot and queries the summary VLM using the *Step Summarization Prompt* to generate a step summary by comparing the before and after screenshots. This summary is then appended to the action history in the format "Step X - Action selected: {action}. {summary}" and provided as context for the next step. The task terminates when the agent outputs a `status` action indicating either "complete" or "infeasible".

---

**System Prompt**

```
You are an agent who can operate an Android phone on behalf of a user.
Based on user's goal/request, you may:
- Answer back if the request/goal is a question
- Complete tasks by performing actions step by step on the phone

Available actions (output in JSON format):
- Status: {"action_type": "status", "goal_status": "complete"/"infeasible"}
- Answer: {"action_type": "answer", "text": "<answer_text>"}
- Click: {"action_type": "click", "element": <description>}
- Long press: {"action_type": "long_press", "element": <description>}
- Input text: {"action_type": "input_text", "text": <text>, "element": <description>}
- Keyboard enter: {"action_type": "keyboard_enter"}
- Navigate home: {"action_type": "navigate_home"}
- Navigate back: {"action_type": "navigate_back"}
- Scroll: {"action_type": "scroll", "direction": <up/down/left/right>,
          "element": <optional description>}
- Open app: {"action_type": "open_app", "app_name": <name>}
- Wait: {"action_type": "wait"}
```

**Guidelines**

```
Here are some useful guidelines you must follow:
General Guidelines:
- Understand the task goal carefully to avoid wrong actions
- Examine the current screenshot carefully; history may be unreliable
- Ensure actions are valid given current observation
- Element descriptions must match what's visible in the screenshot
- Try different description styles if locating fails
- Pick the easiest solution; switch approaches if stuck in failure loops
- Navigate the phone to gather information when needed
- For questions, use "answer" action before "status: complete"
- If desired state is achieved, complete the task immediately

Action Guidelines:
- ALWAYS use "open_app" action to open apps (not app drawer)
- Use "input_text" for typing (don't click keyboard keys one by one)
- Check for default text in fields before typing
- Ensure target elements are visible before clicking/typing
- Explore by scrolling in different directions
- Scroll "down" to view bottom content (opposite to swipe direction)

Text Operation Guidelines:
- To select text: long press → adjust selection pointers → use "select all"
  if needed
- To delete text: use backspace key (can long press to accelerate)
- To copy text: select text → click "copy" in selection bar
- To paste text: long press text box → click "paste" in bar
- Auto-complete dropdown indicates enum field; select from list
```

**Action Selection Prompt**

```
[System Prompt with Role & Action Space]

The current user goal/request is: {goal}

Here is a history of what you have done so far:
{history}

The current screenshot is also given to you.

[Guidelines]

Now output an action from the above list in the correct JSON format,
following the reason why you do that. Your answer should look like:

Output format:
Reason: ...
Action: {"action_type":...}

Your Answer:
```

**Step Summarization Prompt**

```
[System Prompt]

The (overall) user goal/request is: {goal}
Now I want you to summarize the latest step.
You will be given the screenshot before you performed the action (which has a text
    label "before" on the bottom right), the action you chose (together with the
    reason) and the screenshot after the action was performed (A red dot is added
```

```
    to the screenshot if the action involves a target element/position/area,
    showing the located position. Carefully examine whether the red dot is pointing
    to the target element.).

This is the action you picked: {action}
Based on the reason: {reason}

By comparing the two screenshots and the action performed, give a brief summary of
    this step. This summary will be added to action history and used in future
    action selection, so try to include essential information you think that will
    be most useful for future action selections like what you intended to do, why,
    if it worked as expected, if not what might be the reason (be critical, the
    action/reason/locating might be wrong), what should/should not be done next,
    what should be the next step, and so on. Some more rules/tips you should follow:
- Keep it short (better less than 100 words) and in a single line
- Some actions (like 'answer', 'wait') don't involve screen change, you can just
    assume they work as expected.
- Given this summary will be added into action history, it can be used as memory to
    include information that needs to be remembered, or shared between different
    apps.
- If the located position is wrong, that is not your fault. You should try using
    another description style for this element next time.

Summary of this step:
```

**Visual Grounding Prompt**

```
Output the bounding box in the image corresponding to the content
"{description}" with grounding.
The output should be only [x1,y1,x2,y2].
```

## E. Future Work

Although VenusBench-Mobile establishes a systematic evaluation framework with capability diagnostics, several promising directions remain to be explored for more holistic GUI agent assessment. We outline three complementary directions for future benchmark development to guide the next generation of GUI agents toward greater adaptability, lifelong learning, and automated scalability.

**Online Learning.** While our benchmark preliminarily evaluates online learning through HTML tasks with randomized button functionalities, this remains rudimentary. Future work should establish online learning as a standalone dimension, evaluating both intra-task adaptation and inter-task self-evolution with richer randomization (e.g., page transition logic, navigation depth).

**Lifelong Assistant.** Current benchmarks evaluate isolated task episodes, while real-world assistants must operate continuously, accumulating knowledge about user preferences and app ecosystems. Future evaluation should span extended periods, assessing agents' long-term memory and personalization capabilities.

**Agent-Simulated Users.** Manual task design limits benchmark scale. Leveraging LLM-based agents to simulate realistic user behaviors could automatically generate diverse scenarios—varied instructions, multi-round dialogues, and environmental noise—enabling scalable benchmark expansion while maintaining quality through automated validation.

*Table 6.* List of VenusBench-Mobile apps and number of tasks for each one.

| App | Description | # Tasks (Primary + Subset) |
|---|---|---|
| *Apps from AndroidWorld* | | |
| Audio Recorder | A sound recording app for capturing and organizing high-quality audio clips. | 9 + 6 |
| Broccoli | A recipe management app for adding, categorizing, and following cooking guides. | 13 + 4 |
| Calendar | A calendar app for scheduling, modifying, and managing daily events and reminders. | 9 + 2 |
| Camera | A photography app for capturing high-resolution photos and recording videos. | 2 + 0 |
| Chrome | A web browser app for searching information and navigating online platforms. | 5 + 2 |
| Clock | A timekeeping app featuring world clocks, alarms, stopwatches, and timers. | 1 + 0 |
| Contacts | A contact management app for storing personal communication information. | 1 + 0 |
| Simple Draw Pro | A creative drawing app for sketching, painting, and saving digital artwork. | 8 + 6 |
| Files | A file explorer app for browsing, moving, and organizing the Android filesystem. | 22 + 6 |
| Simple Gallery Pro | A media viewing app for browsing and managing personal photos and videos. | 5 + 0 |
| Joplin | A note-taking app designed for creating, editing, and syncing encrypted notes. | 8 + 2 |
| Markor | A markdown editor app for creating and organizing plain-text notes and folders. | 30 + 12 |
| OpenTracks | A sports tracking app for recording GPS tracks and monitoring workout statistics. | 1 + 0 |
| OsmAnd | A navigation app providing offline maps and route planning with location markers. | 1 + 0 |
| Pro Expense | An expense tracking app for monitoring personal finances and managing budgets. | 19 + 8 |
| Retro Music | A music player app for organizing and listening to local audio libraries. | 4 + 2 |
| Settings | A system configuration app for managing device preferences. | 8 + 4 |
| SMSMessages | An SMS app for sending, receiving, and managing text messages. | 5 + 0 |
| Tasks | A task management app for tracking to-do lists, deadlines, and priorities. | 5 + 2 |
| VLC | A media player app for playing various video and audio file formats. | 22 + 8 |
| *New apps in VenusBench-Mobile* | | |
| Calculator | A scientific calculator app providing advanced mathematical computing functions. | 2 + 0 |
| CalcYou | A versatile utility app offering basic arithmetic, unit conversion, and function graphing. | 2 + 0 |
| Phone | The default app for making calls. | 3 + 2 |
| Tomato | A productivity timer app implementing the Pomodoro technique for focus management. | 3 + 0 |
| Fitbook | A health tracking app for logging exercises, daily steps, and nutritional intake. | 3 + 0 |
| MediLog | A personal medical app for recording medication schedules and health history. | 5 + 2 |
| ZipXtract | A file management app specialized in compressing and decompressing files in .zip format. | 3 + 0 |

*Table 7.* Action space of VenusBench-Mobile.

| Action | Parameters | Description |
|---|---|---|
| click | x, y | Perform a single tap at the given screen position |
| double_tap | x, y | Execute two rapid taps at the given screen position |
| long_press | x, y | Press and hold at the given screen position |
| drag | start_x, start_y, end_x, end_y | Swipe from the starting position to the ending position |
| input_text | text | Enter text into the currently active input field |
| scroll | start_x, start_y, end_x, end_y, direction | Perform scrolling from the starting position to the ending position in a given direction (up/down/left/right) |
| navigate_home | — | Go back to the device home screen |
| navigate_back | — | Return to the prior screen |
| keyboard_enter | — | Simulate pressing the enter/return key |
| wait | — | Pause execution to allow screen content to load |
| answer | text | Send a textual reply to the user or ask for clarification |
| status | goal_status | Indicate task completion as *success* or *failure* |

*Table 8.* Fine-grained diagnosis of agent capabilities across varying proficiency levels. We report the success rate (%) on distinct dimensions defined in the PUDAM taxonomy: **P**erception, **U**nderstanding, **D**ecision, **A**ction, and **M**emory. Columns are paired to contrast performance on basic (L1-2) vs. advanced (L3-4) tasks, highlighting the critical degradation in high-level Perception and Memory.

| Model | Perception (P) | | Understanding (U) | | Decision (D) | | Action (A) | | Memory (M) | |
|---|---|---|---|---|---|---|---|---|---|---|
| | L1-2 | L3-4 | L1-2 | L3-4 | L1-2 | L3-4 | L1-2 | L3-4 | L1-2 | L3-4 |
| *Open-source* | | | | | | | | | | |
| UI-Venus-72B | 17 | 12 | 20 | 10 | 11 | 18 | 15 | 20 | 20 | 10 |
| UI-Venus-7B | 11 | 2 | 10 | 6 | 11 | 6 | 8 | 7 | 13 | 1 |
| Qwen3-VL-30B-A3B | 10 | 6 | 9 | 9 | 17 | 4 | 8 | 13 | 12 | 4 |
| Qwen3-VL-8B | 9 | 2 | 7 | 6 | 13 | 3 | 7 | 7 | 10 | 3 |
| GUI-Owl-7B | 8 | 4 | 7 | 6 | 13 | 3 | 6 | 13 | 11 | 1 |
| MA3(GUI-Owl-7B) | 12 | 12 | 13 | 10 | 11 | 13 | 12 | 13 | 13 | 10 |
| *Closed-source* | | | | | | | | | | |
| Gemini-3-Pro | 43 | 24 | 37 | 37 | 40 | 35 | 38 | 27 | 41 | 31 |
| GPT-5.1 | 30 | 20 | 28 | 25 | 28 | 26 | 26 | 33 | 27 | 27 |
| **Average** | **17.5** | **10.3** | **16.4** | **13.6** | **18.0** | **13.5** | **15.0** | **16.6** | **18.4** | **10.9** |

