# OpenReview forum: "VenusBench-Mobile: A Challenging and User-Centric Benchmark for Mobile GUI Agents with Capability Diagnostics"
_ICML.cc/2026/Conference — ICML 2026 spotlight_

### Official Review · Reviewer_qtjK · 2026-03-12

**Soundness:** 3
**Presentation:** 3
**Significance:** 3
**Originality:** 3
**Overall Recommendation:** 5
**Confidence:** 3

**Summary:**

The paper introduces VenusBench-Mobile, a benchmark for evaluating mobile GUI agents in realistic, user-centric scenarios. Unlike prior app-centric benchmarks that focus on isolated application functions, VenusBench-Mobile adopts a user-intent–driven task design that reflects how people interact with mobile devices in practice. The benchmark also introduces a capability-oriented diagnostic framework that attributes failures to underlying abilities such as perception, reasoning, and memory, enabling more fine-grained analysis of agent behavior. Experiments on several state-of-the-art agents show that performance drops substantially compared to existing benchmarks, revealing major limitations in perception, memory, and robustness to environmental variations. These results suggest that current mobile GUI agents remain far from reliable real-world deployment and highlight the need for more robust architectures and evaluation frameworks.

**Compliance With Llm Reviewing Policy:**

Affirmed.

**Final Justification:**

The rebuttal addressed my main concerns.

**Key Questions For Authors:**

1. **Validity of MLLM-based evaluation:**
   Many tasks rely on Qwen3-VL-30B-A3B-Instruct as an automatic judge. How do the authors ensure that its judgments are reliable and unbiased? For example, was the judge validated against human annotations or compared with alternative MLLM judges? Evidence of judge reliability would increase confidence in the reported success rates.

2. **Annotation consistency:**
   The PUDAM taxonomy plays a central role in the diagnostic analysis. Could the authors report inter-annotator agreement statistics or other measures of annotation consistency for the capability labels and difficulty levels? This would help validate the reliability of the annotation process.

**Limitations:**

Yes

**Strengths And Weaknesses:**

### Soundness

**Strengths**
- The benchmark is well motivated: unlike prior app-centric benchmarks, this work adopts a user-intent–driven design that better reflects real mobile usage scenarios, including vague instructions and multi-round interactions.
- The PUDAM capability taxonomy enables fine-grained analysis of agent behavior, and the evaluation metrics (success rate, dimension-wise accuracy, stability pass rate) are clearly defined.
- Empirical results reveal substantial performance drops across models and identify key failure modes (particularly perception and memory), while also reporting inference cost for efficiency comparison.

**Weaknesses**
- Many tasks rely on MLLM judges for evaluation (e.g., Qwen3-VL-30B-A3B-Instruct), but the reliability of this automatic judging is not rigorously validated, raising concerns about bias and reproducibility.
- The paper does not report inter-annotator agreement for the PUDAM annotations, leaving some uncertainty about the consistency of capability labeling.

---

### Presentation

**Strengths**
- The paper clearly motivates the need for a user-intent–driven benchmark and positions the work well relative to existing app-centric benchmarks.
- The evaluation section is detailed and provides useful insights into performance gaps and capability bottlenecks.

---

### Significance

**Strengths**
- The benchmark targets an important emerging problem: evaluating mobile GUI agents in realistic usage scenarios rather than isolated app tasks.
- The large performance drop across models highlights a substantial gap between current agent capabilities and real-world deployment, emphasizing the need for stronger evaluation frameworks.

**Weaknesses**
- The stability evaluation uses only 20 base tasks, which may limit the statistical strength of conclusions about robustness under environment variations.

---

### Originality

**Strengths**
- The user-intent–driven task design represents a meaningful shift from traditional app-centric benchmarks.
- The capability-oriented diagnostic taxonomy and stability evaluation provide useful tools for identifying root causes of agent failures.
- The hybrid evaluation pipeline combining programmatic verification with MLLM-based judgment is a practical design for complex GUI tasks.

**Weaknesses**
- Many components build on existing infrastructures (e.g., AndroidWorld, emulator-based environments), so the novelty lies more in task design and evaluation scope than in fundamentally new methodology.

---

> ### Author Rebuttal · Authors · 2026-03-31
>
> We thank the reviewer for the thorough feedback. Below we address each concern.
>
> ---
>
> ## Weaknesses
>
> > Many tasks rely on MLLM judges for evaluation (e.g., Qwen3-VL-30B-A3B-Instruct), but the reliability of this automatic judging is not rigorously validated, raising concerns about bias and reproducibility.
>
> **Response:** We conducted a human validation study on two representative models (one open-source: UI-Venus-72B; one closed-source: Gemini-3-Pro + UI-Venus-72B) covering all 180 MLLM-judged instances (90 tasks × 2 models). Two annotators independently judged each instance while blinded to the MLLM judge's outputs.
>
> The results demonstrate near-perfect agreement across all comparisons. For UI-Venus-72B, all three parties (Annotator A, Annotator B, MLLM judge) produced identical judgments on all 90 tasks (Cohen's κ = 1.00). For Gemini-3-Pro + UI-Venus-72B, Annotator A fully agreed with the MLLM judge (κ = 1.00), while Annotator B disagreed on a single task (κ = 0.97); upon review, this case was attributed to an annotation error by Annotator B. Aggregating across all 180 instances, the human–human agreement is κ = 0.98 (179/180, 99.4%) and the human–MLLM agreement is κ = 0.98–1.00, confirming that the MLLM judge is highly reliable and consistent with expert human judgment.
>
> Furthermore, as detailed in Appendix B.4, each of the seven judge interfaces is equipped with carefully designed test cases (a total of 16 positive and negative illustrative examples across all interfaces). These test cases served as built-in calibration references during prompt development: we iteratively refined the judge prompts until consistent agreement with human judgment was achieved on these calibration instances. We attribute the strong human–MLLM agreement to this rigorous calibration process, which leaves minimal room for subjective interpretation.
>
> ---
>
> > The paper does not report inter-annotator agreement for the PUDAM annotations, leaving some uncertainty about the consistency of capability labeling.
>
> **Response:** Each task's PUDAM capability labels (five dimensions × four levels) were independently annotated by two trained annotators following Appendix A. Disagreements were resolved through discussion and consensus. We additionally conducted an LLM-based annotation pass as a cross-check.
>
> The inter-annotator agreement between the two human annotators is quadratic weighted Cohen's κ = 0.83 (almost perfect agreement), with 97.2% of annotations within ±1 level. The LLM cross-validation achieves κ = 0.70 (substantial agreement) against the consensus labels. Table 7 also provides indirect validation: consistent performance degradation from L1–2 to L3–4 confirms annotations capture increasing complexity.
>
> ---
>
> > The stability evaluation uses only 20 base tasks, which may limit the statistical strength of conclusions about robustness under environment variations.
>
> **Response:** The 20-task subset is a trade-off between rigor and cost: each generates 4 variants, yielding 100 instances per agent. Our seed–variant paradigm makes expansion straightforward once base infrastructure is in place. Despite moderate size, results reveal a clear pattern: near-zero SPR across all agents (Table 4), with the best model at only 15%, constituting strong evidence of agent brittleness regardless of sample size.
>
> ---
>
> > Many components build on existing infrastructures (e.g., AndroidWorld, emulator-based environments), so the novelty lies more in task design and evaluation scope than in fundamentally new methodology.
>
> **Response:** We chose AndroidWorld for comparability (40+ agents on its leaderboard), real-world fidelity, and low migration cost. Beyond task design, we contributed substantial infrastructure extensions: (1) dynamic noise injection for NR tasks (modifying the emulator control layer to inject disruptions mid-execution), (2) multi-round dialogue support for MR tasks (a fundamentally new interaction mode not in original AndroidWorld), and (3) automated SE pipelines for dark-mode, tablet, and translation variants. See `env_extensions/` and `noise_injection/` in our repository.
>
> ---
>
> ## Key Questions For Authors
>
> > How do the authors ensure MLLM judge reliability?
>
> **Response:** See W1 above. Human validation on 180 instances yields human–human κ = 0.98, human–MLLM κ = 0.98–1.00. Each judge includes 16 calibration test cases (Appendix B.4).
>
> ---
>
> > Could the authors report inter-annotator agreement for PUDAM?
>
> **Response:** See W2 above. Quadratic weighted κ = 0.83 (almost perfect) between human annotators, with LLM cross-validation κ = 0.70.

---

> > ### Author Rebuttal · Reviewer_qtjK · 2026-04-04
> >
> > Thanks for the author's response, and it resolved my concerns. I'll keep my current score.

---

### Official Review · Reviewer_XnRH · 2026-03-13

**Soundness:** 3
**Presentation:** 4
**Significance:** 3
**Originality:** 3
**Overall Recommendation:** 5
**Confidence:** 5

**Summary:**

This paper proposes VenusBench-Mobile, a novel GUI navigation online benchmark in the Android environment. Different from the existing online benchmarks, the tasks designed in this benchmark is more fine-grained and more challenging. The user-intent-driven task design can better reflect realistic use cases, and the capability oriented annotation enriches the behavior analysis. In total, 149 tasks are created under different user-intent-driven task categories, and 80 more tasks are designed for agent stability evaluation. The benchmark has been evaluated in a variety of models, and a comprehensive analysis is conducted to demonstrate the impact of this benchmark

**Compliance With Llm Reviewing Policy:**

Affirmed.

**Final Justification:**

The rebuttal addressed my main concerns and I will keep the score.

**Key Questions For Authors:**

1. Section 3.2 and Appendix A. How to determine the capability taxonomy of each task? It seems to involve a lot of manual efforts, and the boundary between each proficiency level is vague.
2. Section 4, Dimension-wise Accuracy. It is possible that in some cases, only one or two capabilities fail, and the whole task fail. In dimension wise accuracy, it seems to be inaccurate to mark all the capabilities of the failed cases as fail. Is it possible to use VLM as a judge to determine the failure mode in a more fine grained way?

**Limitations:**

The author may consider improving the demension wise accuracy calculation to make it more fine grained.

**Strengths And Weaknesses:**

Strength:
1. User-intent-driven task design and capability oriented annotation provides a fine-grained benchmark for GUI agent evaluation and analysis.
2. Extra stability evaluation can help testing the GUI agents robustness across different task variants
3. The paper provides comprehensive details of the benchmarks, including task categories and examples, detailed prompts as the judge, etc.

Weakness:
The benchmark creation process involves a lot of manual efforts, which may not be very scalable. It would be great if more tasks can be included with the same granurity.

---

> ### Author Rebuttal · Authors · 2026-03-31
>
> We thank the reviewer for the constructive feedback. Below we address each concern.
>
> ---
>
> ## Weakness
>
> > The benchmark creation process involves a lot of manual efforts, which may not be very scalable. It would be great if more tasks can be included with the same granurity.
>
> **Response:** The manual effort is intentional and necessary for the 149 primary tasks. Each serves as a seed atomic task targeting specific PUDAM dimensions and difficulty levels, requiring expert knowledge to ensure correctness of initialization logic, verification functions, and capability annotations. Automated generation cannot yet achieve this rigor, especially for CF, GSA, and GUIM where task validity depends on subtle environmental constraints.
>
> However, scalable expansion on top of atomic tasks is straightforward. Variant tasks can be generated through systematic perturbations, as demonstrated by our SE subset (4 variants per base task). We can also leverage LLMs for instruction paraphrasing and compositional combinations. This seed–variant paradigm anchors quality in human-crafted seeds while scaling quantity efficiently.
>
> ---
>
> ## Key Questions For Authors
>
> > Section 3.2 and Appendix A. How to determine the capability taxonomy of each task? It seems to involve a lot of manual efforts, and the boundary between each proficiency level is vague.
>
> **Response:**  We provide clarification on the design rationale, the level definitions, and the annotation procedure of the PUDAM taxonomy.
>
> **Why five dimensions.** PUDAM was derived from decomposing the GUI agent execution loop, informed by recent works such as UI-Venus (Gu et al., 2025) and UI-TARS (Qin et al., 2025). A pure-vision GUI agent processes each step: it *perceives* the screenshot, *understands* the instruction, makes a *decision*, executes an *action* on the GUI, and *memorizes* relevant information across steps. These five orthogonal dimensions capture distinct aspects of agent competence.
>
> **Why four levels.** Inspired by SAE autonomous driving levels: L1 (basic atomic operations), L2 (deterministic multi-step execution), L3 (dynamic adaptation under uncertainty), L4 (reflective autonomous behavior). Definitions and examples for every level–dimension pair are in Appendix A.
>
> **Annotation procedure.** Once the capability taxonomy is defined, we annotate each task with its PUDAM scores, specifying the minimum proficiency level required in each dimension for successful completion. This annotation was independently performed by two trained annotators following the definitions in Appendix A, with disagreements resolved through discussion. We additionally conducted an LLM-based annotation pass as a cross-check. The inter-annotator agreement between the two human annotators yields a quadratic weighted Cohen's κ = 0.83 (raw agreement 83.0%), indicating almost perfect agreement according to the Landis & Koch scale. Furthermore, 97.2% of all annotations fall within ±1 level, confirming that the rare disagreements are minor boundary cases. The LLM-based cross-check achieves κ = 0.70 (substantial agreement) against the final consensus labels. Given the strong human–human agreement, each task's final PUDAM labels are determined by the two human annotators' consensus, with the LLM pass serving as supplementary verification.
>
> ---
>
> > Section 4, Dimension-wise Accuracy. It is possible that in some cases, only one or two capabilities fail, and the whole task fail. In dimension wise accuracy, it seems to be inaccurate to mark all the capabilities of the failed cases as fail. Is it possible to use VLM as a judge to determine the failure mode in a more fine grained way?
>
> **Response:** Our Acc_dim uses credit-based logic: successful tasks earn credits per dimension proportional to required levels; failed tasks earn none. By aggregating credits across all tasks, we identify relative strengths and bottlenecks per dimension. This is a task-level metric that is intentionally conservative—failure is attributed to all required dimensions even though only one or two may be the actual bottleneck.
>
> A finer-grained approach would require a step-level verifier or PRM analyzing execution trajectories, which constitutes significant future work. That said, our metric already reveals meaningful patterns: consistent degradation from L1–2 to L3–4 (Table 7), Memory as the dominant bottleneck, and clear model-level differences. These provide a reliable lower bound that would remain valid under finer-grained diagnosis.
>
> ---
>
> > Limitations: The author may consider improving the demension wise accuracy calculation to make it more fine grained.
>
> **Response:** We agree with this suggestion. As discussed above, we acknowledge the limitation of the current task-level attribution approach and will include a more explicit discussion in the Limitations section of the revised manuscript, along with concrete directions for building step-level diagnostic verifiers.

---

> > ### Author Rebuttal · Reviewer_XnRH · 2026-04-04
> >
> > Thank you for answering the questions. I will keep the score. Good luck with the submission!

---

### Official Review · Reviewer_fBbn · 2026-03-14

**Soundness:** 3
**Presentation:** 4
**Significance:** 4
**Originality:** 4
**Overall Recommendation:** 5
**Confidence:** 5

**Summary:**

This paper proposes a online benchmark for evaluating general-purpose mobile GUI agents, named VenusBench-Mobile. It focuses on two important directions: (1) what to evaluate; and (2) how to evaluate. Experiments demonstrate that even SOTA GUI Agents still achieve low performances on VenusBench-Mobile.

**Compliance With Llm Reviewing Policy:**

Affirmed.

**Final Justification:**

The rebuttal addresses my concerns. I will keep the score.

**Key Questions For Authors:**

The benchmark is named as VenusBench-Mobile. I am wondering that are the authors planning to release a series of benchmarks, like VenusBench-Desktop, VenusBench-Web?

**Limitations:**

Yes

**Strengths And Weaknesses:**

# Strength
1. The paper is well-written and the direction is very promising
2. I really like the focus of the paper: (1) what to eval and (2) how to eval. It touches the keys of building intelligent GUI Agents, since current evaluations are far from sufficient to lead the trend.
3. The proposed benchmark is meaningful. Experiments show obvious gaps.

# Weakness:
1. The evaluated models are insufficient. It would be better to provide results on some leading models, like Kimi K2.5, Claude-series, UI-Tars/Seed-series. And for these closed-source models, could you provide the results on single model, instead of composed framework.

---

> ### Author Rebuttal · Authors · 2026-03-31
>
> We sincerely thank the reviewer for the thoughtful and constructive feedback, which has been invaluable in improving our manuscript. We are encouraged by the reviewer's recognition of our benchmark's focus on *what to evaluate* and *how to evaluate*, as well as the significance of the performance gaps revealed by our experiments. Below, we provide detailed point-to-point responses to each concern.
>
> ---
>
> ## Weakness
>
> > The evaluated models are insufficient. It would be better to provide results on some leading models, like Kimi K2.5, Claude-series, UI-Tars/Seed-series. And for these closed-source models, could you provide the results on single model, instead of composed framework.
>
> **Response:** We thank the reviewer for the valuable suggestion. We agree that broader model coverage strengthens the benchmark and have taken additional steps to address this concern.
> We have added evaluation results for **Kimi K2.5** and **Seed 2.0**, both evaluated in an **end-to-end manner as standalone agents** rather than within a composed planner–executor framework. Specifically, we developed new agent classes that interface directly with each model's native API, enabling single-model evaluation without relying on an external grounding executor. The results are as follows:
>
> | Model | FA | CF | VA | MR | GSA | GUIM | HGB | NR | BC | Total |
> |---|---|---|---|---|---|---|---|---|---|---|
> | ***From original manuscript (Table 3)*** | | | | | | | | | | |
> | UI-Venus-72B *(best open-source)* | 22.7 | 4.6 | 12.5 | 0.0 | 10.0 | 0.0 | 17.7 | 50.0 | 0.0 | 15.4 |
> | Gemini-3-Pro + UI-Venus-72B *(best closed-source)* | 54.6 | 4.6 | 56.3 | 20.0 | 0.0 | 11.1 | 41.2 | 75.0 | 40.0 | 36.9 |
> | ***Newly added (end-to-end, standalone agent)*** | | | | | | | | | | |
> | Kimi K2.5 | 40.9 | 0.0 | 50.0 | 10.0 | 0.0 | 0.0 | 20.6 | 43.8 | 50.0 | 24.8 |
> | Seed 2.0 | 18.2 | 4.6 | 43.8 | 10.0 | 0.0 | 11.1 | 17.7 | 31.3 | 20.0 | 18.1 |
>
> The results further highlight the challenging nature of VenusBench-Mobile. Even recent competitive models such as Kimi K2.5 and Seed 2.0 achieve relatively low overall performance (24.8% and 18.1%, respectively), with consistently limited success on several capability dimensions (e.g., CF, GSA, GUIM). These observations are consistent with our main findings regarding the difficulty of realistic, user-intent–driven GUI tasks.
> Both models will be added to Table 3 in the final manuscript with full per-category breakdowns.
> However, the stability evaluation and the evaluation of additional frontier models such as Claude-series and UI-Tars are not yet finished within the rebuttal period due to time constraints. We will add those in the final manuscript.
>
> ---
>
> ## Key Questions For Authors
>
> > The benchmark is named as VenusBench-Mobile. I am wondering that are the authors planning to release a series of benchmarks, like VenusBench-Desktop, VenusBench-Web?
>
> **Response:** We thank the reviewer for this question. We have the plans to extend the VenusBench series to desktop and web environments. The core design principles of VenusBench-Mobile including the task design and capability taxonomy are intended to be generalizable across platforms.
> When extending to new evaluation scenarios, we will also explore proposing new design concepts and capability dimensions that address the unique challenges of desktop and web GUI environments (e.g., richer layout structures, cross-tab workflows).
>
> ---
>
> We hope the above responses thoroughly address the reviewer's concerns. We are deeply grateful for the insightful suggestions, which have significantly helped us enhance the quality and completeness of our work. Should there be any remaining questions, we would be happy to provide further clarification.

---

> > ### Author Rebuttal · Reviewer_fBbn · 2026-04-04
> >
> > Thanks for your reply. I will keep my score.

---

### Decision · Program_Chairs · 2026-04-30

**Decision:**

Accept (spotlight)

**Comment:**

Reviewers agree that VenusBench-Mobile introduces a valuable test for mobile GUI agents.
The benchmark focuses on realistic user tasks instead of simple app functions.
It also adds a capability taxonomy to find out exactly why agents fail.
The initial evaluation shows that current models still struggle with real world tasks [fBbn, qtjK].

While the main idea is strong, reviewers raised a few questions about the setup. Reviewer fBbn wanted to see results from more recent models [fBbn]. Reviewers XnRH and qtjK asked about the reliability of the automatic judges and the human labels [XnRH, qtjK]. They also worried about the amount of manual work needed to build the tasks [XnRH].

In the rebuttal, the authors shared new test results for models like Kimi 2.5 and Seed 2.0. They also provided clear numbers to prove that human graders and automatic judges agree almost perfectly.
In addition, they explained how their base tasks can be easily expanded to create more tests.

The reviewers found these answers convincing. All three reviewers kept their positive scores. The benchmark provides a solid tool for the mobile agent field. Therefore, the recommendation is to accept the submission.